# Foreshock properties illuminate nucleation processes of slow and fast laboratory earthquakes

David C. Bolton [1,5] ✉, Chris Marone[2,3], Demian Saffer[1] & Daniel T. Trugman[4]

Understanding the connection between seismic activity and the earthquake nucleation process is a fundamental goal in earthquake seismology with important implications for earthquake early warning systems and forecasting. We use high-resolution acoustic emission (AE) waveform measurements from laboratory stick-slip experiments that span a spectrum of slow to fast slip rates to probe spatiotemporal properties of laboratory foreshocks and nucleation processes. We measure waveform similarity and pairwise differential travel-times (DTT) between AEs throughout the seismic cycle. AEs broadcasted prior to slow labquakes have small DTT and high waveform similarity relative to fast labquakes. We show that during slow stick-slip, the fault never fully locks, and waveform similarity and pairwise differential travel times do not evolve throughout the seismic cycle. In contrast, fast laboratory earthquakes are preceded by a rapid increase in waveform similarity late in the seismic cycle and a reduction in differential travel times, indicating that AEs begin to coalesce as the fault slip velocity increases leading up to failure. These observations point to key differences in the nucleation process of slow and fast labquakes and suggest that the spatiotemporal evolution of laboratory foreshocks is linked to fault slip velocity.

Elucidating the physics of earthquake nucleation is a fundamental goal in earthquake seismology and is key for improving earthquake early warning systems and understanding foreshock sequences and earthquake interaction. Foreshocks, which are small earthquakes that precede the mainshock, are thought to be a signature of the nucleation process, and thus their properties and evolution may provide insights into the impending mainshock[1–11]. However, foreshocks are not a common feature of all tectonic mainshocks, which limits their utility as a robust precursory signature of earthquake failure[12,13].

In contrast to tectonic earthquakes, laboratory earthquakes are often preceded by robust foreshock activity, in the form of pre-seismic acoustic emissions (AEs), over a wide range of conditions[14–32]. AE activity typically occurs throughout all stages of the seismic cycle, but often accelerates rapidly as the fault slip rate begins to increase as loading transitions to become partly inelastic leading up to failure[14,18,19,23,27,28,31]. In addition, laboratory seismicity typically exhibits Gutenberg-Richter frequency-magnitude scaling that evolves systematically throughout the seismic cycle[14–17,20–22,24,28,33]. AE properties are thought to be modulated by several factors, including fault zone heterogeneity, shear stress, fault roughness, fault zone dilation, fault slip rate, and fault orientation[17–19,23,27,28,32,34].

Despite this rich body of work, it is currently unknown if laboratory foreshock sequences delineate a geometric fault zone structure that evolves into catastrophic failure. It is well known that localization of shear deformation in granular fault zones is a key ingredient for modulating frictional behavior[35,36]. However, evidence of such

[1]University of Texas Institute for Geophysics, Jackson School of Geosciences, University of Texas, Austin, TX, USA. [2]Department of Geosciences, Pennsylvania State University, University Park, PA, USA. [3]Departimento di Scienze della Terra, La Sapienza Universita di Roma, Rome, Italy. [4]Nevada Seismological Laboratory, University of Nevada, Reno, NV, USA. [5]Present address: Bureau of Economic Geology, Jackson School of Geosciences, University of Texas, Austin, TX, USA. ✉e-mail: chasbolton19@gmail.com

localization processes preceding stick-slip instabilities in experimental fault zones is sparse, with most existing data focused on fault or fracture formation within intact rocks[37-40]. Part of the problem is that imaging the spatiotemporal properties of AEs requires precise event locations with uncertainties of a few mm or less. Inverting for AE locations, particularly in a gouge fault zone of finite width, is challenging due to non-trivial ray geometries, sub-optimal sensor coverage, and large uncertainties in phase arrival times[29].

Here, we circumvent these issues by quantifying waveform similarity via cross-correlation, and by measuring pair-wise differential travel times (DTT) between event pairs to track relative changes in event locations[41,42]. Waveform similarity measurements have proven to be a useful tool for building high-resolution catalogs, understanding repeating earthquakes, and aftershock sequences along tectonic fault zones[43-47]. Here we measure waveform similarity and DTT from acoustic emission data recorded throughout hundreds of laboratory seismic cycles to study the evolution of foreshocks. In our experiments, we systematically varied the fault zone normal stress to produce a spectrum of stick-slip instabilities[48-50]. The fastest stick-slip instabilities at normal stresses of 14 MPa show a significant increase in waveform similarity and a reduction in DTT once the shear stress surpasses ~80%

of the peak frictional strength. In contrast, waveform similarity and DTT remain constant throughout the slowest stick-slip instabilities at 8 MPa. Together, the reduction in DTT and increase in waveform similarity for fast stick-slip events is indicative of a transition from pervasive to localized deformation in the lead-up to failure. The spatiotemporal evolution of foreshocks in our experiments is modulated by pre-seismic fault slip and is broadly consistent with theoretical models of nucleation where aseismic creep is an intrinsic part of the nucleation process[1,2,51,52].

## Results

### Stick-slip properties

By systematically modulating the normal stress, we produced a range of stick-slip instabilities with peak fault slip velocities ranging from 20–1000 μm/s and stress drops from 0.1–1 MPa (Fig. 1b). We classify slip events as slow or fast depending on the peak fault slip velocities reached during their co-seismic slip phase (see inset to Fig. 1a, b). Slip events at 8 and 10 MPa have peak fault slip rates <500 μm/s and are classified as "slow". The stick-slip instabilities at 12 and -14 MPa have peak slip rates >=500 μm/s and are considered "fast". Higher normal stresses produce events with larger stress drops and faster peak slip velocities, consistent with previous laboratory studies and theoretical arguments[48,50,53-55].

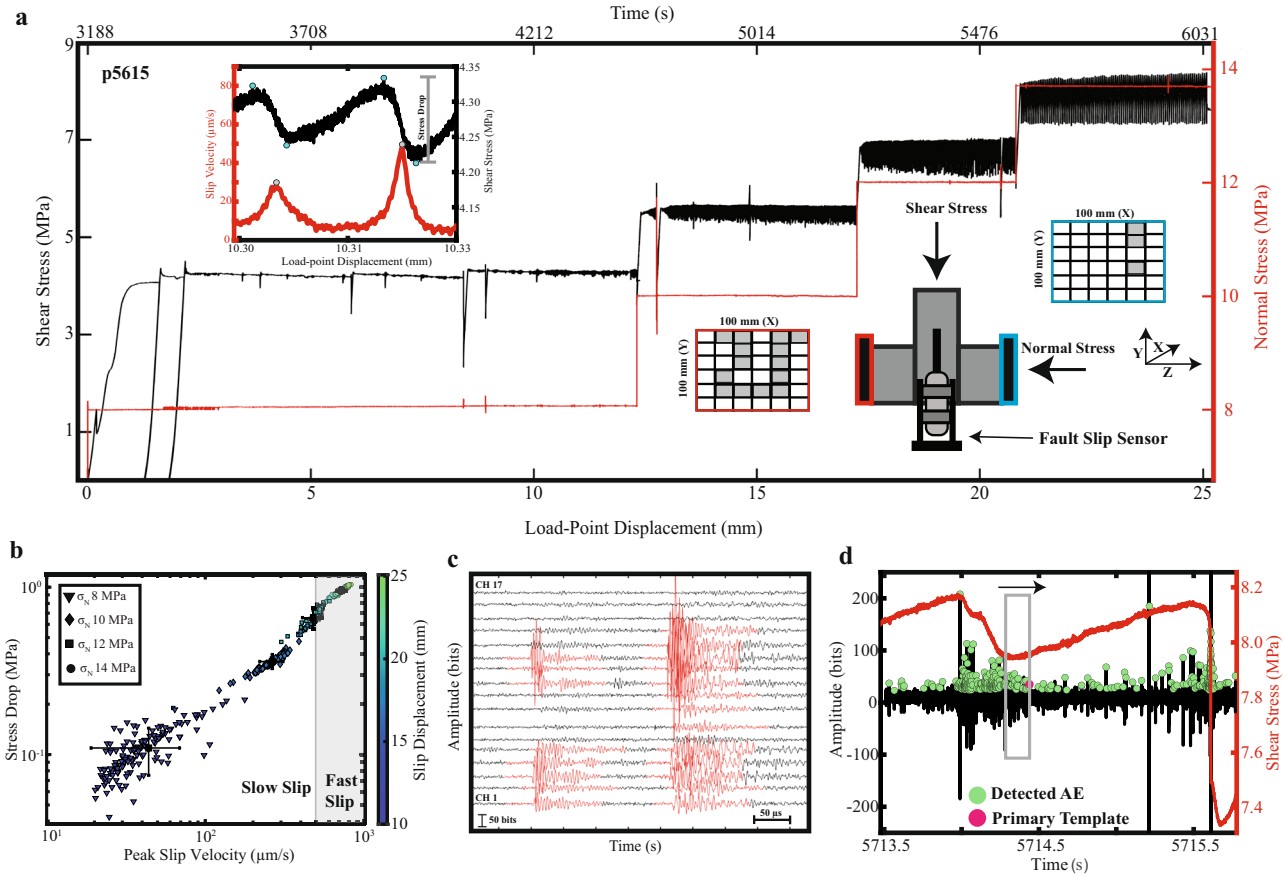

**Fig. 1 | Experimental setup and evolution of laboratory earthquakes. a** Shear stress and normal stress plotted as a function of load-point displacement for Experiment p5615. Two unload-reload cycles were performed after shearing ~2–3 mm. Slow stick-slip instabilities begin to nucleate after shearing ~8 mm. Inset on bottom right shows double-direct shear configuration with acoustic blocks. Acoustic blocks are 100 × 100 mm² and placed adjacent to the granite forcing blocks; inset on bottom right shows acoustic blocks with gray shading to represent stations used for AE monitoring. Inset on top left shows slip velocity and shear stress evolution for a representative slow slip event at 8 MPa. Blue circles denote the peaks and troughs of the co-seismic slip phase and are used to estimate stress drops. Gray circles represent peak slip velocities reached during co-seismic slip.

**b** Stress drop as a function of peak slip velocity for stick-slip cycles in (**a**). For each normal stress mean values are plotted with black symbols and error bars represent +−1 standard deviation. Data points are color coded according to cumulative slip displacement. **c** Continuous acoustic emission data for all 17 stations. Highlighted in red are 150 μs templates extracted using our event detection procedure (see "Methods"). **d** Continuous AE signal for channel 2 (black) along with the detected AEs shown in green and shear stress in red. We compute waveform similarity and differential travel-times using a sliding window throughout the seismic cycle (depicted with gray box). For each sliding window, we compute waveform similarity and DTT between a primary template (the last AE in the window) and every event bounded within the window.

## Evolution of waveform similarity and differential travel-times throughout the seismic cycle

We measure waveform similarity by cross-correlating AE templates as a function of time using a moving window approach (Figs. 1d, 2); see "Methods" section for details regarding our AE detection procedure and template creation. We set the length of each moving window to 5% of the recurrence interval of the seismic cycle and allow each window to overlap the previous window by 90%. The algorithm proceeds as follows: (1) find all templates that are temporally bounded within the moving window, (2) set the final template as the primary template of interest and cross-correlate all templates with respect to this primary template, (3) measure waveform similarity (see "Methods" section below for details), and (4) record the maximum cross-correlation coefficient within the moving window.

We focus on waveform pairs that have high waveform similarity because they can provide insights into event locations and source properties[41,42,44–47]. AE waveforms are a convolution of the event source,

propagation effects, and sensor/recording response. Hence, if two events are co-located, have similar source mechanisms, and are recorded on the same instrument then they will produce similar waveforms. Because our cross-correlation procedure compares events that are recorded on the same station, differences in waveform similarity are driven by differences in source properties and/or in source locations.

We complement our waveform similarity measurements by tracking relative changes in AE locations with DTT (Fig. 3). We focus on relative changes in AE locations rather than absolute locations because we are interested in understanding how AEs evolve throughout the seismic cycle with respect to one another. Furthermore, absolute locations often have large uncertainties, which could easily mask subtle spatiotemporal trends that are likely important for understanding nucleation processes, but which are resolvable in relative locations[41,42]. For example, we located the AEs shown in Fig. 3 by minimizing a robust L1-norm between the observed arrival time data and predicted travel time data. According to the absolute locations, the two events have an

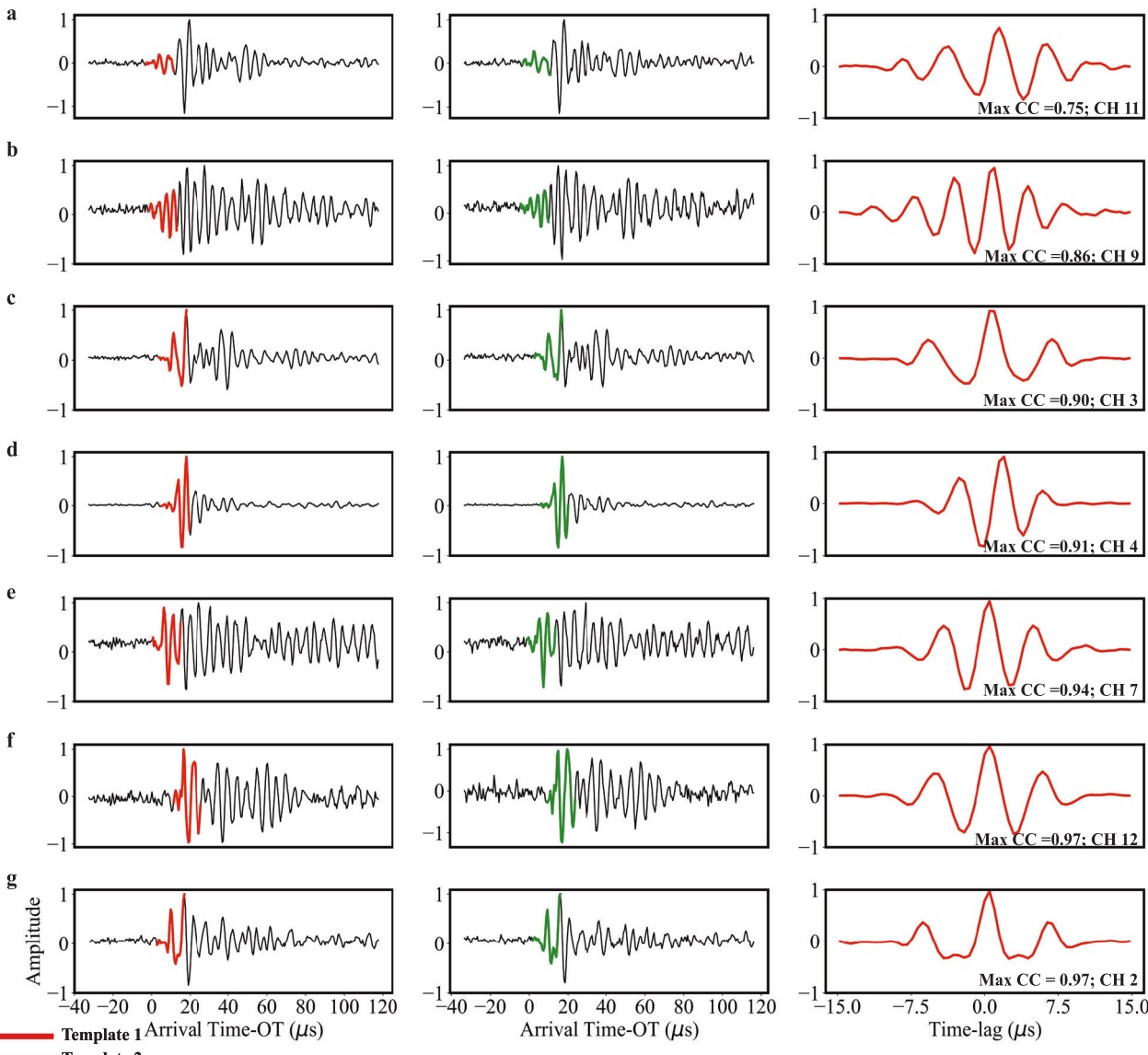

**Fig. 2 | Waveform similarity measurements. a–g** Representative example of template waveforms used to compute waveform similarity. Note, each row corresponds to a different channel. The entire 150 µs waveforms are shown in black and templates used for cross-correlating are shown in red (template 1) and green (template 2). Templates are 15 µs in length and start 5 µs before the p-arrival and

extend 10 µs after the p-arrival. The last column in each panel shows the cross-correlation function derived from cross-correlating the red and green templates. Note, plots are arranged vertically according to similarity. We quantify waveform similarity by measuring the maximum of the absolute value of the cross-correlation function and average the top 6 channels.

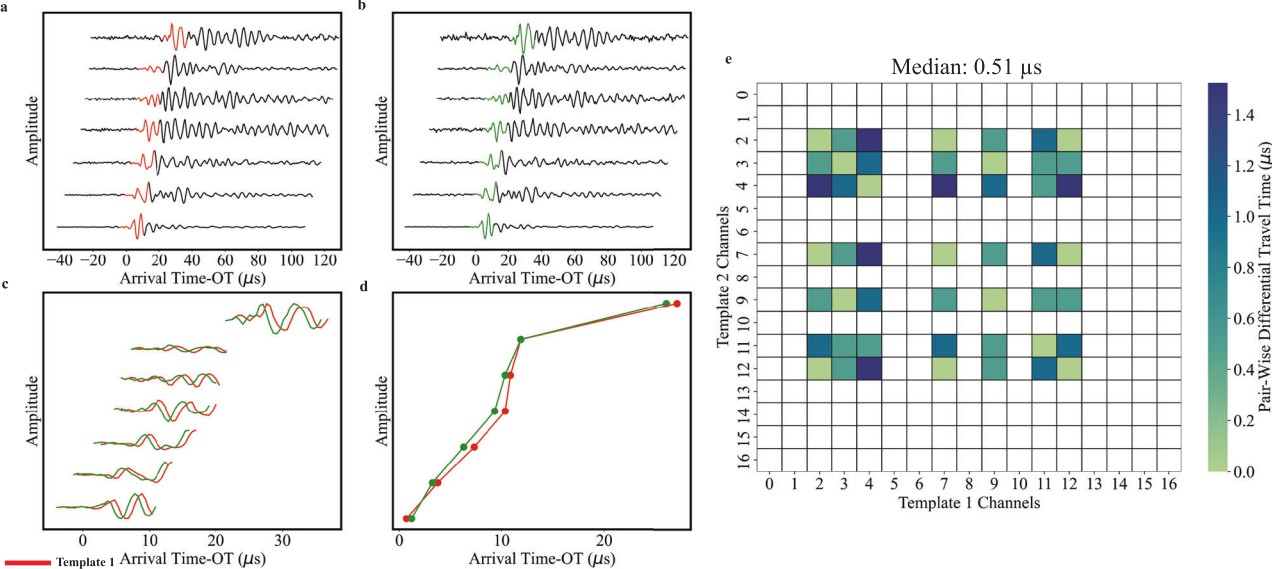

**Fig. 3 | Pair-wise differential travel-time measurements. a, b** Waveforms and associated templates from Fig. 2. We locate each event by using a grid search approach, where the optimal solution minimizes an L1-norm between the arrival time data and a set of theoretical travel-times. Waveforms are offset vertically for clarity and plotted as a function of time after removing the origin time (OT) from the p-arrival. The time difference from T = 0 until the p-arrival denotes the theoretical travel-time. **c, d** Arrival time moveouts for the two events shown in (**a**) and (**b**), respectively. Note, the moveouts are practically identical, indicating that the events are co-located. **e** Pair-wise differential travel-times between templates shown in (**a**) and (**b**). Each grid node represents the differential travel time between template 1 and template 2, for a given station pair (see Eq. 1 in main text for details). We quantify the differential travel time between two events my measuring the median of the upper half of the differential travel time matrix shown in (**e**); the differential travel time between template 1 and template 2 is 0.51 µs (<3 mm; assuming $V_p$ = 5500 m/s).

inter-event distance of 5 mm (Fig. 3), which is a significant fraction of the fault dimension (~10% of the fault length). However, the DTT between the two events is ~0.5 µs, which corresponds to ~2.8 mm (i.e., $V_p$ of 5500 m/s). By focusing on relative changes in DTT measurements, we obtain information about the relative location/distance between event pairs without having to invert arrival time data, and thus avoid the large uncertainties associated with absolute locations. If two events are located near one another, then they produce similar arrival time moveout vectors and the difference between these vectors is small (Fig. 3). DTT are therefore proxies for the relative locations/distances between two events and tracking DTT throughout the seismic cycle can provide insights into how AEs evolve in space relative to one another. For example, a decrease in DTT as a function of time implies that AEs are beginning to nucleate closer together, while an increase in DTT as a function of time indicates that the AEs are moving farther apart.

We measure DTT between all event pairs and a primary template of interest (last event in window) for each moving window (Fig. 3; see "Methods" section). For each event pair, we report the median of the upper half of the DTT matrix (i.e., Fig. 3e) and time stamp each moving window with the minimum DTT for all event pairs. We focus on the minimum DTT for each moving window because we are interested in observing localization effects throughout the seismic cycle. In other words, if the events are coalescing in space, then the minimum DTT should decrease throughout the seismic cycle.

We plot the maximum cross-correlation coefficient and minimum DTT returned from our moving window as a function of stress state (Fig. 4). Because we are interested in probing nucleation processes, we focus only on data that occur during the inter-seismic period (defined as the time from the minimum shear stress to the peak stress of a given seismic cycle). For each moving window, we normalize the instantaneous shear stress by the peak shear stress reached before co-seismic failure. The small gray circles in Fig. 4 represent raw measurements and we quantify these data points by measuring their median using a sliding window approach (Figs. 4, S1). Waveform similarity and DTT are essentially constant throughout the seismic cycle for slow labquakes at 8 MPa normal stress (Fig. 4a). The slip events at 10 and 12 MPa show a modest decrease in waveform similarity over the first ~half of the inter-seismic loading period, followed by a slight increase once the fault reaches ~60% of its peak stress (Fig. S1B, C). The DTT track this behavior, showing a modest increase followed by a slight decrease once the fault surpasses ~50–60% of its peak stress. The fastest events at 14 MPa show a similar behavior to the events at 10–12 MPa, however, the temporal variations in waveform similarity and DTT are much stronger. Waveform similarity reduces from 0.85 to 0.75 during the early stages of the seismic cycle (shear stress <40% of the peak stress), remains constant, and then increases rapidly from ~0.75 to 0.85 once the fault surpasses ~80% of its peak stress (Fig. 4b). Again, DTT tracks this behavior, increasing during the early stages of the seismic cycle, and then decreasing rapidly once the fault reaches ~80% of its peak stress.

It is also worth noting that the median waveform similarity decreases and DTT increases with increasing normal stress. For example, between 40 and 60% of the peak stress at 8 MPa, the median cross-correlation values fluctuate around 0.87, while the DTT are between 1 and 2 µs. In contrast, at 14 MPa waveform similarity fluctuates around 0.75 and DTT are between 7.5 and 10 µs (Figs. 4, S1).

In addition to the waveform properties, the fault shear stress and slip rate coevolve differently for slow and fast slip instabilities (Fig. S2). During slow events, the fault does not fully lock during the inter-seismic period. At 8 MPa the fault continues to creep throughout the inter-seismic period and reaches a minimum slip rate of ~5–7 µm/s, before accelerating during the later stages of the seismic cycle (Fig. S2). In contrast, during fast instabilities the fault is nearly locked during the inter-seismic period and reaches a minimum slip rate of ~0–3 µm/s before accelerating during the nucleation phase. Here, we define the nucleation stage as the time period during which the fault slip velocity increases slowly from a background rate (e.g., locking) to a higher slip rate reached at the peak stress (Fig. 1 inset).

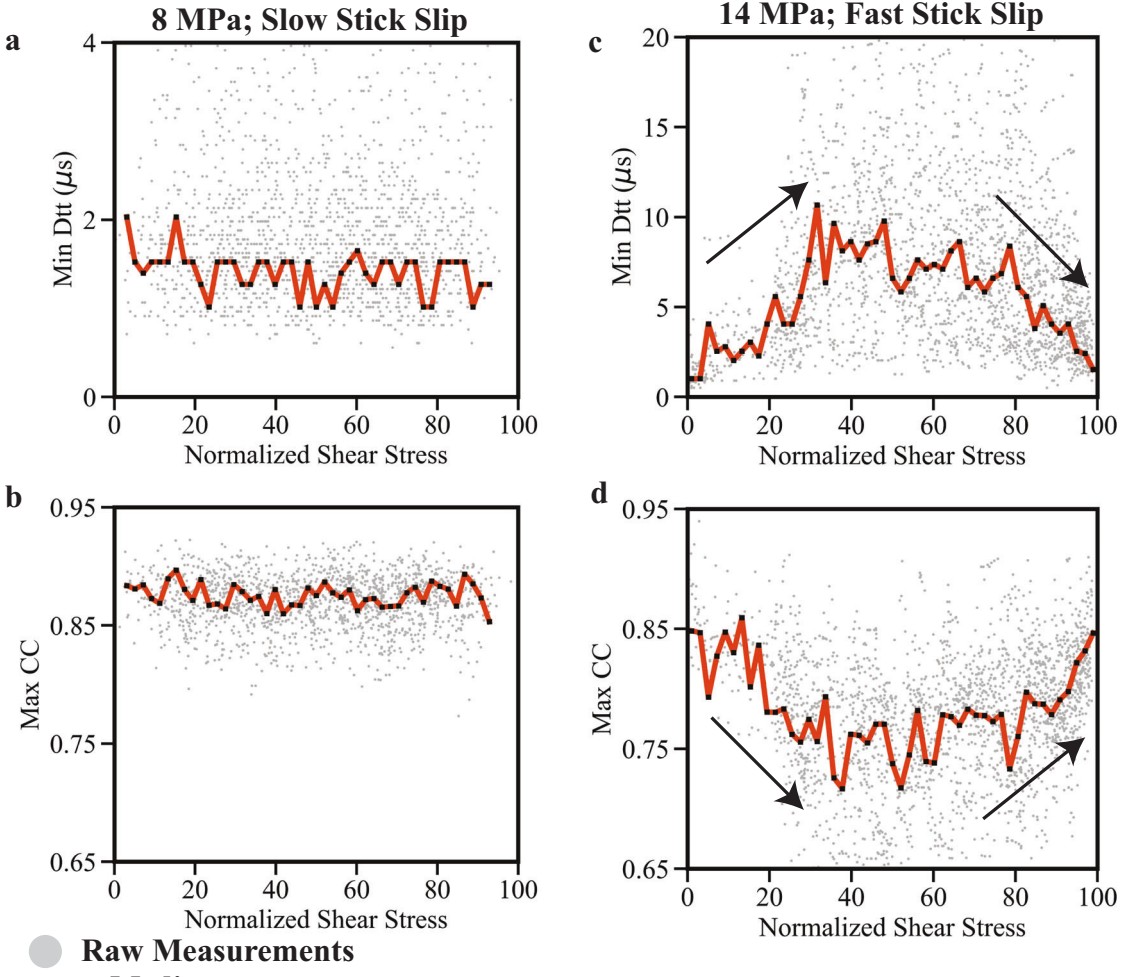

**Fig. 4 | Evolution of differential travel times and waveform similarity throughout the seismic cycle. a–d** Maximum cross-correlation coefficient and minimum DTT as a function of peak stress reached prior to co-seismic failure. Percent of peak stress is calculated by normalizing the instantaneous shear stress by the peak shear stress reached prior to co-seismic failure. We cross-correlate waveforms and estimate DTT using AE data from 15 seismic cycles (Fig. S1). Gray dots represent raw measurements and red line denotes the median derived from a moving window approach. DTT and waveform similarity remain constant throughout the seismic cycle for slow slip events at 8 MPa. Fast stick-slip events at 14 MPa show notable increase in DTT and decrease in similarity when the shear stress is below 40% of its peak stress. In contrast, the data show a significant uptick in waveform similarity and a decrease in DTT once the fault surpasses ~80% of the peak stress.

## Discussion

Earthquakes that rupture the same fault patch and have the same source mechanism will produce waveforms with high-similarity[45,46]. For tectonic faults, such events are referred to as repeating earthquakes if the waveforms exhibit high similarity (e.g., cross-correlation coefficients >=0.95)[44,45]. The same concepts are applicable to laboratory seismic data, and while we may not have true (strictly defined) repeaters, AEs with high waveform similarity and low DTT are evidence of foreshock localization. Events that occur close together in space have small DTT, and by tracking changes in DTT and waveform similarity during the seismic cycle we can monitor relative changes in event locations. We integrate such waveform data with fault slip rate measurements in Fig. 5 and propose a micromechanical model that explains the temporal changes in waveform similarity and DTT of Fig. 4.

We envision that AEs generated during the inter-seismic period are a byproduct of frictional slip along grain contact junctions. Because our experiments are conducted using granite surfaces coated with a thin layer of powder, AEs are likely generated from the following micromechanical processes: (i) the failure of rock-rock asperities or (ii)

slip at contact junctions formed within fault gouge[53,55,56] (Fig. 5). Postmortem analysis of previous experiments shows that most of the shearing interface is covered by a thin layer of quartz powder and only a small percentage of the surface contains regions of rock-rock asperities[53]. Hence, it is likely that most of the acoustic radiation emanates from contact junctions within localized regions (~mm size areas) of the gouge layer[18,19,30,57,58] (Fig. 5).

For slow stick-slip events, the fault creeps throughout the entire inter-seismic period, and waveform similarity and DTT remain constant throughout the seismic cycle (Fig. 4). The fact that DTT remains constant throughout the inter-seismic period indicates that AEs are not evolving systematically in space. However, AEs that nucleate during slow events have lower DTT and higher waveform similarity throughout the seismic cycle compared to AEs that nucleate during fast stick-slip events (Figs. 4 and S1). Evidence of a localization process should be apparent in the DTT measurements because they provide information about the relative distances between event pairs, and thus, smaller DTT are indicative of AEs that are more tightly clustered in space. As the fault approaches failure, these localized regions along the fault zone do not evolve (i.e., become wider and/or smaller), as is apparent

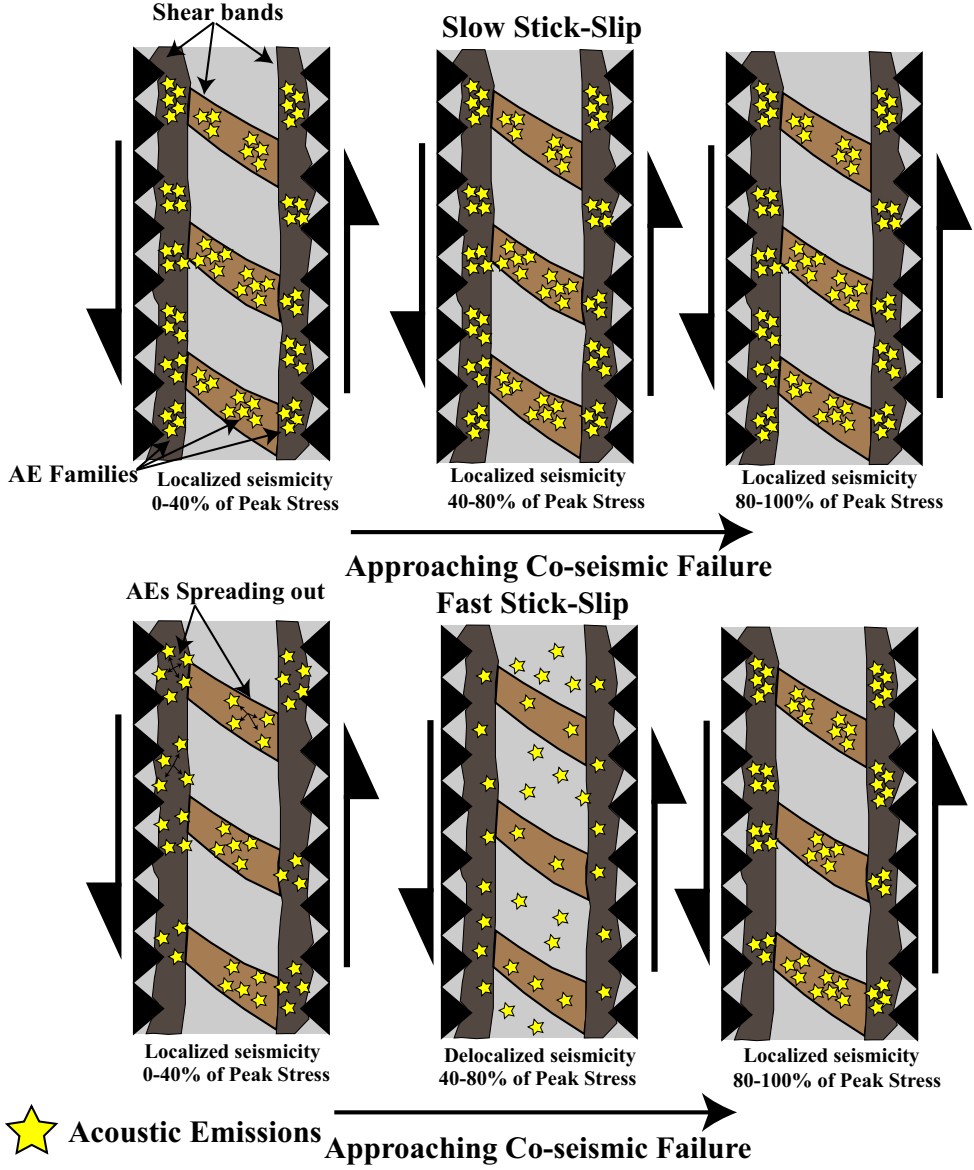

**Fig. 5 | A micromechanical model describing the spatiotemporal evolution of AEs and foreshocks throughout the laboratory seismic cycle.** We propose that AEs in our experiments are generated from slip along localized planes of weakness within the gouge layer which are depicted as brown parallel and sub-parallel structures that span the fault zone. Yellow stars are used to denote individual AE events and groups of spatially clustered AEs represent families. Motivated by the data in Fig. 4 we depict the spatiotemporal evolution of AEs at three locations within the seismic cycle. The inter-seismic period of slow events consists of multiple AE families with high AE similarity and low DTT. In addition, waveform similarity and DTT remain constant throughout the seismic cycle. We depict these characteristics by showing multiple AE families that are tightly clustered/localized in space and invariant to position within the seismic cycle. In contrast, fast stick-slip events in Fig. 4 show three distinct regimes throughout the seismic cycle. The first regime occurs when the shear stress is below 40% of its peak and is characterized by decreasing waveform similarity and increasing DTT. In this regime AEs are spreading out across the fault zone and transitioning from localized to delocalized/randomly distributed. DTT is the highest and waveform similarity is the lowest between 40 and 80% of the peak stress (i.e., second regime), indicating that AEs are located far from one another and spread out across the fault zone. Once the fault surpasses 80% of its peak shear strength, DTT begins to decrease and waveform similarity begins to increase, indicating that AEs are transitioning from being delocalized/randomly distributed to more localized and clustered in space. We depict this localization process with multiple groups of tightly clustered AEs. It is worth noting that the clustering and localization of AEs prior to fast stick-slip (between 80 and 100% of the peaks stress) is similar to the localization/clustering that occurs throughout all stages of the seismic cycle for slow slip events.

from the time invariant DTT and waveform similarity measurements (Figs. 4, S1). Furthermore, the fault slip velocity remains high throughout the inter-seismic period, reaching a minimum slip rate between 5 and 7 μm/s (-38−52% of the far-field loading rate) (Fig. S2). Together the fault slip velocity, DTT, and waveform similarity indicate that inter-seismic fault creep is correlated with localized shear deformation. This observation is also consistent with the interpretation that the fault creeps throughout the inter-seismic period—and thus does not lock up or slip slowly enough to allow fault healing and accumulation of elastic strain energy, a set of conditions that promotes the occurrence of slow slip[59−62].

On the other hand, for fast stick-slip events DTT increases and waveform similarity decreases when the fault is below 40% of its peak frictional strength. Once the fault surpasses -80% of its peak strength DTT decreases and waveform similarity increases as the fault transitions from locked to creeping (Fig. 4). We propose that this evolution is indicative of a transition from pervasive to localized shear deformation as the fault approaches failure (Fig. 5). In the early stages of the seismic

cycle, DTT increases and waveform similarity decreases, indicating that the AEs are spreading out in space. Once the fault approaches failure DTT decreases and waveform similarity increases, indicating that AEs are becoming closer in space. The fact that the fault does not creep until it reaches ~80% of its peak stress for fast stick-slip events provides an explanation for the abrupt increase in waveform similarity and decrease in DTT. Furthermore, this observation is consistent with the low DTT, high similarity, and high inter-seismic fault slip rates at 8 MPa. Low DTT and high waveform similarity appear to be strongly correlated with high slip velocities and are indicators of localized deformation. This interpretation is consistent with microstructural observations of shear localization in granular fault zones, with additional insights into how strain might localize across the seismic cycle[56,63]. Our results are also in agreement with previous laboratory studies that show that AEs migrate from a broader damage zone towards the fault core as failure approaches[31].

Our data support the idea that fault slip velocity plays an important role in modulating AE properties[18,27,28,31]. Furthermore, the data shown in Figs. 4 and S1 support a general relationship between waveform similarity, DTT, and inter-seismic slip velocity. Specifically, waveform similarity scales systematically with the inter-seismic fault slip rate, while DTT scales inversely with the inter-seismic slip rate. For example, waveform similarity is the highest and DTT are the lowest at 8 MPa, where the inter-seismic slip velocities are the largest (Fig. 4). High waveform similarity and low DTT seem to be fingerprints of high inter-seismic fault slip rates. This interpretation is also consistent with the rapid increase in waveform similarity and reduction in DTT prior to fast instabilities.

AEs with low DTT and high waveform similarity could represent the failure of an ensemble of tiny mm to sub-mm fault patches or the failure of one large fault patch[30]. Discriminating between the two is impossible based solely on the data in Fig. 4 because both scenarios would result in low DTT and high waveform similarity. One way to discriminate between the two cases is to identify families of AEs based upon their similarity. If one assumes that variations in waveform similarity are driven by differences in AE locations, then the number of families can be used as a proxy for the number of locations. This assumption neglects the possibility that co-located events could have different source mechanisms, and thus, produce distinct waveforms and belong to different families. However, because our data show a strong inverse relationship between waveform similarity and DTT we assume that differences in locations are the dominant factor that controls waveform similarity (Fig. 4). Nevertheless, we caution the reader that the clustering results and interpretation hinge on this assumption.

To test this idea, we cross-correlated all AE pairs within the inter-seismic period of individual stick-slip cycles and input these data into a Hierarchical Agglomerative Clustering (HAC) algorithm to identify families of AEs (Figs. S3–S4). We used a distance matrix of $1-C_{i,j}$, where $C_{i,j}$ denotes the median waveform similarity between events $i$ and $j$ and merged clusters using a complete linkage. We imposed a distance threshold of 0.4, allowing clusters to contain waveforms with pair-wise cross-correlation coefficients between 0.6 and 1. Using this definition, HAC identified between 23 and 36 families of AEs for slow slip cycles at 8 MPa and between 12 and 15 families of AEs for fast slip cycles at 14 MPa. It is important to note that the number of clusters is strongly affected by the distance threshold, and thus, the results in Figs. S3–S4 should be used with caution. Increasing the threshold above 0.4 is not ideal as it would allow for dissimilar waveforms with cross-correlation coefficients <0.5 to be a part of the same cluster. On the other hand, decreasing the threshold would increase the number of families and would set a stricter definition for what constitutes a family. Thus, the number of clusters identified at a threshold of 0.4 likely represents a lower bound on the total number of potential clusters/families.

Assuming that the number of families is a proxy for the number of locations, the data in Figs. S3 and S4 indicate that there are an ensemble of fault patches failing throughout the inter-seismic period for both slow and fast stick-slip events. Slow slip events produce more locations throughout the inter-seismic period than fast events. This is consistent with the fact that the fault never locks up during the inter-seismic period for slow slip events, allowing AEs to nucleate from multiple regions across the fault plane. It therefore follows that during slow stick-slip AEs are likely broadcasted from an ensemble of localized regions across the entire fault zone, which in turn, produces low DTT and high waveform similarity between event pairs (Fig. 5). These observations are consistent with the time invariant DTT and waveform similarity measurements in Fig. 4 and the idea that AEs are not evolving systematically across the fault. In contrast, fast stick-slip events produce fewer families and localized regions of deformation throughout the seismic cycle. Fast stick-slip events also show a modest increase in the number of families as co-seismic failure approaches which could be explained by the late and rapid increase in fault slip rate (Figs. S2 and S4). Broadly speaking, these results are consistent with the late reduction in DTT and increase in waveform similarity prior to failure (Fig. 4).

In the laboratory, acoustic emissions are the result of micromechanical processes acting along structures with length scales on the order of a few mm and waveforms are enriched with high-frequency energy between 10 and 500 kHz[55,57,58,64,65]. In contrast, tectonic earthquakes occur on much larger and more heterogenous faults in which radiated energy is dominantly <10 Hz for moderate and large events[66,67]. However, by integrating high-resolution measurements of fault zone displacements and seismic radiation, lab experiments provide key insights into the underlying physics that control spatiotemporal patterns in seismicity on earthquake faults.

Our data suggest that pre-seismic AEs (i.e., lab foreshocks) with high waveform similarity and low DTT are fingerprints of fault slip and laboratory nucleation. Our observations are consistent with a pre-slip model of earthquake nucleation, whereby dynamic fault rupture is preceded by a slow creeping front[1,4–9,51]. It is important to note that we do not directly measure the rupture front and/or the associated nucleation zone in our experiments[32,51,52,68]. Thus, we use the word nucleation loosely to denote the time period prior to co-seismic failure, when the fault slip rate is slowly increasing. We refer to AEs during this period as foreshocks (Fig. 1b). In the pre-slip model, it is thought that foreshocks are a manifestation of fault creep[1–10,51]. Our AE and fault zone measurements support this view and are consistent with previous lab studies[27,28,31]. However, it is also possible that some AEs trigger one another, and thus we cannot rule out a cascade triggering model[51,69,70] or other hybrid models[51,71] that connect these two endmembers.

The interpretation that foreshocks and AEs arise from inter-seismic fault creep in the lead-up to failure is broadly consistent with recent work showing that earthquakes along the San Andreas Fault (SAF) are closely linked to local fault creep measured geodetically[72]. Clustered seismicity is correlated with patches that are locked and thus experience much lower or near-zero inter-seismic creep rates. Thus, most of the strain energy is released co-seismically during large M6+ earthquakes. For example, some of the largest events that nucleated along the SAF, such as the 1989 M6.9 and 2004 M6.0 earthquakes, occurred in regions that experience low creep rates[72]. These observations are consistent with our data and proposed micromechanical model (Fig. 5), in which the inter-seismic slip velocity is low for fast stick-slip events, allowing for greater frictional healing and leading to larger stick-slip events (Figs. 1 and S2). In addition, the rapid increase in waveform similarity and reduction in DTT once the fault reaches ~80% of the peak stress suggests that seismicity is localized in both space and time immediately before failure. The localization and spatial clustering observed in Fig. 4 is also consistent with the localization and coalescence of seismicity prior to large earthquakes along the SAF[73].

Systematic changes in pre-seismic fault zone and seismic properties support the notion that earthquakes are preceded by a

preparatory stage, consistent with theoretical models of earthquake nucleation[1,4,51]. Our work builds on previous studies and demonstrates that waveform similarity and DTT can be a useful tool for probing earthquake nucleation and foreshock properties. Fast laboratory earthquakes are preceded by an abrupt increase in fault slip rate, which in turn, broadcasts AEs that coalesce in space and time prior to failure. The lab results are in good agreement with several field studies that show an increase and coalescence of foreshocks/seismic activity prior to large crustal earthquakes[5–11,69–71,73]. Furthermore, there is a growing body of evidence that suggests that fault slip rate plays a fundamental role regulating foreshock properties at the laboratory scale[18,19,27,28,31]. Our work supports this view and further implies that high waveform similarity and low DTT are fingerprints of pre-seismic fault slip and could be a useful tool for tracking changes in fault slip rate and precursory processes along tectonic fault zones.

## Methods

### Stick-slip experiments and acoustic emission measurements

We report on data from 500+ laboratory earthquakes featuring high-resolution measurements of the laboratory fault zone and acoustic properties (Fig. 1). We used a double-direct shear (DDS) configuration to shear experimental fault zones that were 50 mm × 50 mm in area and composed of Westerly granite surfaces roughened with 60-grit and coated with a thin layer (~500 μm) of quartz powder to simulate the wear associated with fault offset. Each fault surface of the DDS was coated with <1 g of quartz powder (Min-U-Sil 40, median grain size of 10.5 μm). The experiment began by first applying an 8 MPa normal stress to the fault, followed by driving the center block at a constant load point displacement rate of 13 μm/s. Once the fault reached its peak strength, we performed two unload-reload cycles in which the shear stress was completely removed and reapplied (Fig. 1a). This procedure facilitates the development of shear fabric and the onset of unstable sliding[35,36].

A horizontal piston applied and maintained a constant normal stress to the sample, while the vertical piston sheared the centered block at constant loading-rate of 13 μm/s throughout the experiment (Fig. 1a). Normal and shear loads were measured with strain-gauge load cells mounted in series with the horizontal and vertical rams, respectively. Fault displacements, both parallel and perpendicular to shear, were measured with direct current displacement transducers (DCDT) mounted to the nose of the pistons and referenced to the load-frame. Fault displacement was measured by mounting a DCDT directly to the bottom of the center block and referenced to the base of the load-frame. Fault displacements and stresses were measured continuously throughout the experiment at 1 kHz using a 24-bit resolution data acquisition system.

We produced a spectrum of stick-slip modes, ranging from slow to fast slip events by altering the stiffness of the system ($k$) and systematically modulating the normal stress[48–50] (Fig. 1). A thin acrylic block was placed in series with the vertical ram to reduce the machine stiffness, allowing the critical stiffness of the fault $k_c$ - $k$, where $k_c$ represents the critical fault weakening rate/stiffness.

Acoustic emission (AE) data were measured continuously at 1.968 MHz throughout the experiment using a 15-bit Verasonics data acquisition system. AE data were measured with broadband (~0.0001–2 MHz) piezoceramic sensors, with a maximum sensitivity between 100 and 500 kHz[55]. We recorded AE data from an array of 17 stations; 14 stations were positioned on the left side of the fault and 3 stations were on the right side of the fault (Fig. 1 inset).

### AE templates and waveform similarity measurements

We use an event detection algorithm to scan through the continuous AE data and develop event catalogs[24,25,28] (Fig. 1d). The algorithm is described in detail along with an extensive sensitivity analysis in Bolton et al. (2021). In summary, the event detection algorithm slides through the continuous AE data and detects events based upon a set of empirical thresholding parameters. The algorithm treats each station independently and catalogs the peak amplitudes and their associated time stamps.

In order to measure waveform similarity and track its evolution throughout the seismic cycle, we first extracted waveform templates from our event catalog (Fig. 1d). We created templates by iterating through the catalog using a moving window approach. To ensure that the detected event was not a false positive, we first checked that it was recorded on at least six stations. We then identified the station associated with the earliest peak amplitude and placed a 150 μs time window centered about this time stamp. This allowed us to extract the full waveform across the entire network without a priori information about the arrival times. Associating waveforms to a common source is trivial for our dataset because the inter-event times (~milliseconds) are much larger than the relative arrival-time moveouts (~microseconds) across the network[29]. The 150 μs window length was selected such that it was smaller than the average AE inter-event times and longer than the relative arrival time moveouts across the network. As a last step, we checked that only one event occurred within our 150 μs time window. With the templates in hand, we then identified their arrival times using a kurtosis-based phase picker, PhasePApy, and updated these picks using PhaseNet[29,74,75].

We then measured waveform similarity by cross-correlating waveform pairs across the entire network of stations. For each waveform pair, we measured the maximum amplitude of their cross-correlation function and averaged the top six channels (Fig. 2). Prior to computing the cross-correlation functions, we truncated the original templates to 15 μs, which included 5 μs of data prior to the p-wave arrival time and 10 μs of data following the arrival time. The 15 μs window length allows us to preserve information pertaining to the source of the event, while excluding unwanted information in the waveform coda.

### Differential travel-time measurements

We measure differential travel-times for all combinations of channel pairs between two template waveforms. More specifically, we define DTT as:

$$\text{DTT} = abs\left(\left(\text{AT}_i^M - \text{AT}_j^M\right) - \left(\text{AT}_i^N - \text{AT}_j^N\right)\right) \quad (1)$$

where $\text{AT}_i^M$ is the arrival time of template $M$ at channel $i$ and $\text{AT}_j^M$ is the arrival time of template $M$ at channel $j$. Similarly, $\text{AT}_i^N$ represents the arrival time of template $N$ at channel $i$ and $\text{AT}_j^N$ represents the arrival time of template $N$ at channel $j$. Note, the arrival time differences for each template are calculated for all channel pairs, resulting in a $K \times K$ matrix, where $K$ represents the number of channels (Fig. 3e). This matrix is symmetric, and the diagonal terms are zero; off diagonal entries in Fig. 3e that are white represent channel pairs that do not have arrival time data.

## Data availability

All data used in this study were collected at The Pennsylvania State University Rock and Sediment Mechanics Laboratory. The processed mechanical (shear stress, normal stress, load-point displacement, etc.) data and waveform templates are publicly available at https://scholarsphere.psu.edu/resources/3e5b02e2-04e8-430c-a0cd-e245ad7363da.

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

## Acknowledgements

Discussions with Ian McBrearty, Shrisharan Shreedharan, Samson Marty, and technical assistance in the laboratory from Steve Swavely are gratefully acknowledged. The material is based upon work supported by the National Science Foundation (NSF) under award number EAR-PF 2050006 to D.C.B. D.C.B. also acknowledges support from the Bureau of Economic Geology at UT-Austin. We also acknowledge support from the European Research Council Advanced Grant 835012 (TECTONIC) and US Department of Energy grants DE-SC0020512 and DE-EE0008763 to C.J.M. and from NSF Award EAR-2121666 to D.T.T.

## Author contributions

D.C.B., D.T.T., and D.S. devised the study. D.C.B. conducted the experiments with assistance from C.M. All authors helped with data analysis and writing the manuscript.

## Competing interests

The authors declare no competing interests.
