## [Peer Review File · Nature Communications]

Foreshock properties illuminate nucleation processes of slow and fast laboratory earthquakesREVIEWER COMMENTS

Reviewer #1 (Remarks to the Author):

This study probes the fault nucleation process by monitoring AE activity during laboratory stick-slip experiments. The authors found that similar AEs increased significantly before fast slip events, and moderately increased before slow slip events. The authors further argued that the systematic evolution of such similar foreshocks before stick-slip event is a by-product of the slow nucleation process. This is nice work and has significance to study the precursor process of fault instability from the similarity of laboratory foreshocks.

Major comments:

1) Methodologically, the authors state that they can achieve slow and fast slip by adjusting the stiffness and normal stress of the experimental machine. This study does not seem to adjust the stiffness, please confirm, and indicate the specific stiffness value.

2) The conclusion of this paper comes from only one experiment, and the normal stress is increased to 8, 10, 12, 14 MPa sequentially. Under each normal stress, there are more than 100 stick-slip events, and the total displacement reaches more than 20 mm. Effects such as smoothing, loss and redistribution of gauge cannot be ignored. In fact, Fig. 1 has shown that stress drop increased clearly with increasing cumulative displacement, indicating the peak slip velocity also increased significantly. This is a very interesting phenomenon and warrants some careful analysis.

3) In seismology, similar earthquakes that can indicate pre-slip are resulted repeated failures of the same asperity. However, the similar AEs claimed in this study have relative distances up to 11 mm, which is larger than the AE crack size ($<<1\text{mm}$?) and the maximum grain size of Westerly granite. It is not robust to conclude that the similar events are repeaters, since successive failures of adjacent asperities can also produce AEs of similar waveforms. There are a total of 17 AE sensors, so the template AE (as shown in Figure 1D with clear P first motion in multiple channels) can be located with sufficient precision. The evolution of AE hypocenter distribution can better explain the nucleation process of fault instability. According to previous research results, fault stick-slip has an evolutionary process from localized nucleation and quasi-static acceleration to dynamic rupturing. This process can be confirmed from AE hypocenter migration.

4) Shreedharan et al. (2020) estimated the magnitude of the stick-slip events (about -4), I like to see the magnitude (may be relative) of AEs and then analyze the change of b value?

Miner comments:

5) Line 60. The angle of the fault to the axis of the maximum principal stress is also an important factor (Lei et al., Fracturing behaviors of unfavorably oriented faults investigated using an acoustic emission monitor. *Advances in acoustic emission technology*: Springer; 2017. p. 229-40.)

6) Line 76-77. Please note that an earlier study has modified and applied the template-matching and location method to experimental AE data (Lei X et al., An effective method for laboratory acoustic emission detection and location using template matching, *Journal of Rock Mechanics and Geotechnical Engineering*, <https://doi.org/10.1016/j.jrmge.2022.03.010>)

Line 90, Please show roughness of the fault surface, which is important for modeling the effective contact area and estimating size of asperities.

7) Line 126-127. As mentioned in 3), repeated earthquakes are repeated failures of the same asperity driven by surrounding creep. There is no sufficient evidence to say that AEs with similar waveforms are repeaters.

8) Line 303. What is the exact sampling interval?

9) Line 584. Why choose only 15 cycles instead of all? Is the choice of time window arbitrary?

10) Your results are inconsistent with Yamashita et al. (2021), which shows that fewer foreshocks occurred along the less heterogeneous fault (without pre-existing gauge) and were driven by pre-slip; in contrast, more foreshocks with a lower b value occurred along the more heterogeneous fault (with pre-existing gauge) and showed features of cascade-up. Some discussion on this is suggested.

11) Figure 1C. It is recommended to color the data points by the cumulative displacement and use different symbols to indicate the normal stress, so that the cumulative effect from slip history mentioned in 2) will be reflected.

Reviewer #2 (Remarks to the Author):

Key results, summary:

The authors of the study "Foreshock properties illuminate nucleation processes of slow and fast laboratory earthquakes" by

Bolton et al. report from a series of stick-slip instabilities (SSI's) during four different levels of normal stress at a constant shear loading-rate of 13 μ m/s. A double-direct shear apparatus on westerly granite samples with a contact area of 50 x 50 mm was used. The surface area was roughened and coated with quartz powder of different grain size. As in previous studies of this kind, higher normal stresses produce larger stress drops and larger peak slip velocities during SSI's. The authors divide the recorded SSI's of the two lower normal stress levels/slower peak slip rates and the higher normal stress levels/faster peak slip rates into slow and fast slipping SSI's, respectively. The entire experiment lasted for ~50 minutes during which 500+ acoustic emissions (AE's) were recorded which are interpreted as lab earthquakes. The recorded AE's exhibiting cross-correlation coefficients (ccc) above the 75th percentile with respect to the total recorded AE's per normal stress level are interpreted as repeating earthquakes, exhibiting co-location and similar source mechanism.

During lower normal stresses (8, 10MPa) which is equivalent to slow SSI's the laboratory fault is constantly creeping without a distinct nucleation stage, whereby 75th percentile ccc's of recorded AE's remain at a high ccc levels (≥ 0.8) throughout. The authors interpret that the high slip rates, experienced during inter-seismic periods, allows less frictional healing and thus result in smaller stress drops during the co-seismic period and spatially more evenly distributed seismicity.

For fast SSI's at 12/~14MPa normal stress the laboratory fault remains locked until ~80% of the peak shear stress is reached, whereby 75th percentile ccc's are increasing after the ~80% shear stress is reached from ~0.7 to a final value below 0.8. The authors interpret that frictional healing is possible because of low slip velocities during inter-seismic periods and thus leads to larger stress drops in the co-seismic phase of the cycle. In addition, waveform similarity increases (compared to the background similarity on the respective normal stress level) after the ~80% shear stress is reached.

The authors discuss that consistent with previous laboratory and field studies, AE's and repeating AE's are fingerprints for pre-seismic fault creep, fault slip and laboratory nucleation prior to SSI's. With their results the authors would support the pre-slip model but cannot rule out that some AE's trigger one another suggesting the cascade model. They also find consistency between their results and findings on the SAF - clustered seismicity correlates with locked areas, low creep rates and areas where large earthquakes occur, and non-clustered seismicity correlates with areas prone to creep.

The authors conclude that fault slip rates play a fundamental role regulating foreshock properties at the laboratory scale. Fast SSI's are preceded by an abrupt increase in fault slip rate, which broadcast repeating AE's that spatiotemporally coalesce prior to failure. Thus, repeating micro-seismicity during pre-seismic fault slip could be a useful tool for tracking changes in fault slip rate and precursory process along tectonic fault zones.

Concerns with main conclusions:

In my view the authors of the manuscript "Foreshock properties illuminate nucleation processes of slow and fast laboratory earthquakes" reveal some differences in AE foreshock properties from

stick-slip instabilities produced in a double-direct shear apparatus depending on different normal stress levels. The main discussion and conclusion is based on repeating earthquakes which are solely determined by elevated cross-correlation coefficients, and not by location and overlapping source radii, which is in my view not sufficient in order to classify repeating earthquakes (concern 1).

Also, choosing ccc limits ≥ 0.7 for repeating earthquakes (during high normal stresses) is maybe not the best choice (concern 2) as ccc values for repeater identification in literature often exceed 0.95 (e.g., Chaves et al. (2020); Kato et al. (2016); Nadeau and Johnson (1998); Naoi et al. (2015)).

Aside of the fact, that only waveform similarity is considered as a criterion for repeating earthquakes, the authors argue that they are observing an increase in similarity after $\sim 80\%$ of the shear-stress is reached before failure for the fast SSI's, which I can to some extent support when investigating Figure 3. However, in this conclusion you are only considering AE similarities on the respective normal stress level (reference background level). According to your data, statistically, AE's exhibit higher similarities for the slow SSI's (concern 3).

Significance:

The topic of understanding properties of pre-seismic fault slip certainly has very high relevance in the field of earthquake physics. However, as mentioned above, your conclusions based on repeating earthquakes lack robustness. A nice overview on repeating earthquakes was published by Uchida and Bürgmann (2019) you cite this study in your manuscript. In section 2 they nicely state the prerequisites for true repeaters. Another study you also have as a reference in your study is Gao et al. (2021). Their study "Misconception of Waveform Similarity in the Identification of Repeating Earthquakes" is focused on the problem stated above.

Because of the stated concerns, drawing main conclusions in the direction of nucleation models (L. 245-249) are made too soon in my understanding.

Suggestions:

I would suggest the authors to improve spherical coverage of AE sensors for an accurate AE location. I am not very familiar with the double-direct shear apparatus at Pennsylvania State University, but yes, I presume you have had these thoughts as well and maybe for these experiments with no confinement an improved spherical coverage is possible. Also deploying sensors which are calibrated in an absolute sense (McLaskey et al., 2015) would allow you to estimate magnitudes and corner frequencies. I am aware myself and that the authors are stating in their manuscript that event location is difficult (L. 66-70), however, to me, an accurate location is unfortunately almost a pre-requisite for any repeater study.

Once, AE sensor coverage is improved you could apply similar technics as Trugman et al. (2020), or continue with complementing your similarity study with a cluster analysis and refine arrival times per determined cluster. Use a relative location method to more accurately locate potential repeating AE's (e.g., Officer et al. (2022), Dong et al. (2019)). If absolutely calibrated AE sensors are deployed estimate source radii (e.g., Selvadurai (2019)). Unfortunately, the experiment would have to be repeated with improved AE sensor coverage and calibrated sensors.

Attachments:

- comments in article (Article_reviewer_comments.pdf)
- comments to figures (Reviewer_comments_on_figures.pdf)

References:

Chaves, E. J., Schwartz, S. Y., & Abercrombie, R. E. (2020). Repeating earthquakes record fault weakening and healing in areas of megathrust postseismic slip. *Science Advances*, 6(32), eaaz9317. <https://doi.org/10.1126/sciadv.aaz9317> (Science Advances)

Dong, L., Zou, W., Sun, D., Tong, X., Li, X., & Shu, W. (2019). Some developments and new insights for microseismic/acoustic emission source localization. *Shock and Vibration*, 2019.

Gao, D., Kao, H., & Wang, B. (2021). Misconception of Waveform Similarity in the Identification of Repeating Earthquakes. *Geophysical Research Letters*, e2021GL092815. (Geophysical Research Letters)

Kato, A., Fukuda, J. i., Kumazawa, T., & Nakagawa, S. (2016). Accelerated nucleation of the 2014 Iquique, Chile Mw 8.2 earthquake. *Scientific reports*, 6(1), 1-9.

McLaskey, G. C., Lockner, D. A., Kilgore, B. D., & Beeler, N. M. (2015). A robust calibration technique for acoustic emission systems based on momentum transfer from a ball drop. *Bulletin of the Seismological Society of America*, 105(1), 257-271. (Bulletin of the Seismological Society of America)

Nadeau, R. M., & Johnson, L. R. (1998). Seismological studies at Parkfield VI: Moment release rates and estimates of source parameters for small repeating earthquakes. *Bulletin of the Seismological Society of America*, 88(3), 790-814. (Bulletin of the Seismological Society of America)

Naoi, M., Nakatani, M., Igarashi, T., Otsuki, K., Yabe, Y., Kgarume, T., Murakami, O., Masakale, T., Ribeiro, L., & Ward, A. (2015). Unexpectedly frequent occurrence of very small repeating earthquakes ($-5.1 \leq MW \leq -3.6$) in a South African gold mine: implications for monitoring intraplate faults. *Journal of Geophysical Research: Solid Earth*, 120(12), 8478-8493. (Journal of Geophysical Research: Solid Earth)

Officer, T., Zhu, L., Li, Z., Yu, T., Edey, D. R., & Wang, Y. (2022). Application of the double-difference relocation method to acoustic emission events in high-pressure deformation experiments. *Physics and Chemistry of Minerals*, 49(8), 1-13.

Selvadurai, P. A. (2019). Laboratory insight into seismic estimates of energy partitioning during dynamic rupture: An observable scaling breakdown. *Journal of Geophysical Research: Solid Earth*. (Journal of Geophysical Research: Solid Earth)

Trugman, D. T., McBrearty, I. W., Bolton, D. C., Guyer, R. A., Marone, C., & Johnson, P. A. (2020). The Spatiotemporal Evolution of Granular Microslip Precursors to Laboratory Earthquakes. *Geophysical Research Letters*, 47(16), e2020GL088404. <https://doi.org/https://doi.org/10.1029/2020GL088404>

Uchida, N., & Bürgmann, R. (2019). Repeating earthquakes. *Annual Review of Earth and Planetary Sciences*, 47, 305-332. (Annual Review of Earth and Planetary Sciences)

Reviewer #3 (Remarks to the Author):

Review of manuscript NCOMMS-22-30982 submitted to Nature Communications
Foreshock properties illuminate nucleation processes of slow and fast laboratory earthquakes by David C. Bolton, Chris Marone, Demian Saffer, and Daniel T. Trugman

This is a well-presented study of stick slip tests performed on quartz gouge embedded between Westerly granite blocks in a DDS apparatus. The authors modified the normal stress to induce slow and fast slip events and monitored the AE signals during the tests. They then analyzed waveform similarity between events as function of time to failure and combined this with differential travel time analysis in an effort to estimate relative locations of AE events. These methods are well established in the analysis of field seismological data but haven't yet been applied in this way to lab tests with AE monitoring.

The authors find that waveform similarity shows significant increase towards failure for fast slip events but not for slow ones. Increase in waveform similarity is interpreted as indicating formation of localized fault patches at which slip nucleates. Since similar signatures were not observed for slow slip events, the authors posit that slow slips lack a clear nucleation stage.

As stated by the authors the study largely confirms the conceptual view of processes leading to stick slip failure in laboratory tests that has emerged from many previous publications which are mostly cited here. To what extent the contention is justified that a significant difference exists

between processes leading to slow slip versus fast slip events is hard to tell as long as it is just based on the proxies analyzed here. Other studies suggest that event clustering is more affected by fault roughness but may occur prior to both fast and slow slips (Goebel et al., 2017, doi:10.1130/G39147.1).

While the results are generally in line with the current view of the nucleation process (as defined here), the study of temporal changes of AE waveform similarity is new and may well be a very valuable tool in analyzing labquake nucleation. This raises the only question or rather comment I have with this paper: The authors have an array of 17 stations at their disposal (BTW, it would be important to have a clear picture of the sensor geometry in a figure). Not knowing the set up of the sensor array I would assume this could be used to actually locate the events, which in turn would allow using hypocenter cluster analysis in space-time that could then be directly compared with the proposed waveform similarity evolution. In my view this could significantly strengthen the main results/conclusions from this work.

In summary, this clearly is an interesting study, that deserves publishing with some minor revision. Including a comparison with evolution of AE hypocenters correlation and possibly source types would clearly strengthen the paper.

Sincerely
Georg Dresen

Review comments are provided verbatim below in black Times New Roman font.
Our responses are provided in green Times New Roman font.

We include a clean version of the manuscript and one showing track changes.
Line numbers quoted here are for the clean version of the manuscript.

Reviewer #1 (Remarks to the Author):

This study probes the fault nucleation process by monitoring AE activity during laboratory stick-slip experiments. The authors found that similar AEs increased significantly before fast slip events, and moderately increased before slow slip events. The authors further argued that the systematic evolution of such similar foreshocks before stick-slip event is a by-product of the slow nucleation process. This is nice work and has significance to study the precursor process of fault instability from the similarity of laboratory foreshocks.

Major comments:

1) Methodologically, the authors state that they can achieve slow and fast slip by adjusting the stiffness and normal stress of the experimental machine. This study does not seem to adjust the stiffness, please confirm, and indicate the specific stiffness value.

Yes, we modulate the stiffness of the apparatus by placing an acrylic block in series with the vertical ram and central block of the double-direct assembly. See Figure 1A and lines 309-313 in methods section. The stiffness evolves as a function of slip displacement (see Leeman et al., 2016; 2018). At 8 MPa between 1-2 mm the loading stiffness is ~ 0.002 (1/ μm). See also Shreedharan et al., 2020.

2) The conclusion of this paper comes from only one experiment, and the normal stress is increased to 8, 10, 12, 14 MPa sequentially. Under each normal stress, there are more than 100 stick-slip events, and the total displacement reaches more than 20 mm. Effects such as smoothing, loss and redistribution of gauge cannot be ignored. In fact, Fig. 1 has shown that stress drop increased clearly with increasing cumulative displacement, indicating the peak slip velocity also increased significantly. This is a very interesting phenomenon and warrants some careful analysis.

This is a good point. To be clear, the experiments are conducted on Westerly granite blocks roughened with 60-grit and coated with a thin layer of Min-U-Sil 40 (quartz powder with median grain size of 10.5 μm) (see lines 289-295). The effect of slip displacement/shear strain is most significant during the initial stages of shearing, where the development of shear fabric plays an important role in modulating the frictional behavior and the transition from velocity strengthening to weakening (see Marone, 1998; Scuderi et al., 2017). Our analysis is conducted in the latter stages of the experiment when the fault zone has reached a steady-state shear fabric.

Furthermore, our previous works have demonstrated that AE statistics are unaffected by cumulative slip displacement (Riviere et al., 2018; Hulbert et al., 2019; Bolton et al., 2020; 2021). The magnitude of the stick-slip events does indeed become progressively bigger and faster at 8 MPa with increasing slip. However, the effect of cumulative slip displacement on stress drops and event statistics is negligible for stick-slip events between 10-14 MPa (Figure 1A; see also Leeman et al., 2016; Scuderi et al., 2016).

3) In seismology, similar earthquakes that can indicate pre-slip are resulted repeated failures of the same asperity. However, the similar AEs claimed in this study have relative distances up to 11 mm, which is larger than the AE crack size ($\ll 1$ mm?) and the maximum grain size of Westerly granite. It is not robust to conclude that the similar events are repeaters, since successive failures of adjacent asperities can also produce AEs of similar waveforms. There are a total of 17 AE sensors, so the template AE (as shown in Figure 1D with clear P first motion in multiple channels) can be located with sufficient precision. The evolution of AE hypocenter distribution can better explain the nucleation process of fault instability. According to previous research results, fault stick-slip has an evolutionary process from localized nucleation and quasi-static acceleration to dynamic rupturing. This process can be confirmed from AE hypocenter migration.

Yes, this is a good point. We decided not to refer to our events as repeaters. Deriving absolute locations for our system and dataset is complicated because of the small sample size and small errors in arrival time data map to large uncertainties in event locations (e.g., 1 μ s maps to ~ 5 mm). Thus, we use pair-wise differential arrival times between event pairs which gives high resolution relative locations between events. A reduction in pair-wise differential travel time indicates that events are occurring closer together in space. With the new analysis we now show that event locations become more clustered in space as failure approaches. This is especially evident for our higher normal stress experiments, from 10-14 MPa. The fastest events, at 14 MPa, show a rapid reduction in differential travel times, and thus, a stronger degree of localization as failure approaches. This is indeed indicative of localization process that proceeds failure and further supports our waveform similarity measurements. See discussion section for more details.

4) Shreedharan et al. (2020) estimated the magnitude of the stick-slip events (about -4), I like to see the magnitude (may be relative) of AEs and then analyze the change of b value?

Thanks for the suggestion. However, tracking the temporal evolution of b-value throughout the laboratory seismic cycle is not only beyond the scope of our current study, but is also a topic we have previously explored (e.g., Riviere et al., 2018 and Bolton et al., 2021).

Miner comments:

5) Line 60. The angle of the fault to the axis of the maximum principal stress is also an important factor (Lei et al., Fracturing behaviors of unfavorably oriented faults investigated using an acoustic emission monitor. *Advances in acoustic emission technology*: Springer; 2017. p. 229-40.)

Thank you. We now cite this paper in the revised text.

6) Line 76-77. Please note that an earlier study has modified and applied the template-matching and location method to experimental AE data (Lei X et al., An effective method for laboratory acoustic emission detection and location using template matching, Journal of Rock Mechanics and Geotechnical Engineering, <https://doi.org/10.1016/j.jrmge.2022.03.010>)

Thank you. We now cite this paper in the revised text.

Line 90, Please show roughness of the fault surface, which is important for modeling the effective contact area and estimating size of asperities.

We refer the reviewer and readers to Shreedharan et al. 2020 Figure S1 for documentation of the surfaces.

7) Line 126-127. As mentioned in 3), repeated earthquakes are repeated failures of the same asperity driven by surrounding creep. There is no sufficient evidence to say that AEs with similar waveforms are repeaters.

We have removed this statement.

8) Line 303. What is the exact sampling interval?

1.968 MHz

9) Line 584. Why choose only 15 cycles instead of all? Is the choice of time window arbitrary?

The choice of 15 slip cycles is arbitrary. We wanted to analyze enough slip cycles to demonstrate that the trends we observed were robust. As mentioned above, the mechanical and seismic attributes of the seismic cycle are roughly constant once the fault reaches steady state.

10) Your results are inconsistent with Yamashita et al. (2021), which shows that fewer foreshocks occurred along the less heterogeneous fault (without pre-existing gauge) and were driven by pre-slip; in contrast, more foreshocks with a lower b value occurred along the more heterogeneous fault (with pre-existing gauge) and showed features of cascade-up. Some discussion on this is suggested.

This is an interesting observation, but the data presented in our manuscript are insufficient to directly compare with Yamashita et al., 2021. We do not have a comprehensive list of experiments conducted with different gouge thicknesses that can be compared with Yamashita et al. However, in previous works we have addressed the role of particle size and fault zone thickness on AE statistics (Bolton et al., 2020; 2021). As you note, those works show that thicker gouge layers produce fewer and smaller AE.

11) Figure 1C. It is recommended to color the data points by the cumulative displacement and

use different symbols to indicate the normal stress, so that the cumulative effect from slip history mentioned in 2) will be reflected.

Done

Reviewer #2 (Remarks to the Author):

Key results, summary:

The authors of the study “Foreshock properties illuminate nucleation processes of slow and fast laboratory earthquakes” by Bolton et al. report from a series of stick-slip instabilities (SSI's) during four different levels of normal stress at a constant shear loading-rate of 13 μ m/s. A double-direct shear apparatus on westerly granite samples with a contact area of 50 x 50 mm was used. The surface area was roughened and coated with quartz powder of different grain size. As in previous studies of this kind, higher normal stresses produce larger stress drops and larger peak slip velocities during SSI's. The authors divide the recorded SSI's of the two lower normal stress levels/slower peak slip rates and the higher normal stress levels/faster peak slip rates into slow and fast slipping SSI's, respectively. The entire experiment lasted for ~50 minutes during which 500+ acoustic emissions (AE's) were recorded which are interpreted as lab earthquakes. The recorded AE's exhibiting cross-correlation coefficients (ccc) above the 75th percentile with respect to the total recorded AE's per normal stress level are interpreted as repeating earthquakes, exhibiting co-location and similar source mechanism.

During lower normal stresses (8, 10MPa) which is equivalent to slow SSI's the laboratory fault is constantly creeping without a distinct nucleation stage, whereby 75th percentile ccc's of recorded AE's remain at a high ccc levels (≥ 0.8) throughout. The authors interpret that the high slip rates, experienced during inter-seismic periods, allows less frictional healing and thus result in smaller stress drops during the co-seismic period and spatially more evenly distributed seismicity.

For fast SSI's at 12/~14MPa normal stress the laboratory fault remains locked until ~80% of the peak shear stress is reached, whereby 75th percentile ccc's are increasing after the ~80% shear stress is reached from ~0.7 to a final value below 0.8. The authors interpret that frictional healing is possible because of low slip velocities during inter-seismic periods and thus leads to larger stress drops in the co-seismic phase of the cycle. In addition, waveform similarity increases (compared to the background similarity on the respective normal stress level) after the ~80% shear stress is reached.

The authors discuss that consistent with previous laboratory and field studies, AE's and repeating AE's are fingerprints for pre-seismic fault creep, fault slip and laboratory nucleation prior to SSI's. With their results the authors would support the pre-slip model but cannot rule out that some AE's trigger one another suggesting the cascade model. They also find consistency between their results and findings on the SAF - clustered seismicity correlates with locked areas, low creep rates and areas where large earthquakes occur, and non-clustered seismicity correlates with areas prone to creep.

The authors conclude that fault slip rates play a fundamental role regulating foreshock properties

at the laboratory scale. Fast SSI's are preceded by an abrupt increase in fault slip rate, which broadcast repeating AE's that spatiotemporally coalesce prior to failure. Thus, repeating micro-seismicity during pre-seismic fault slip could be a useful tool for tracking changes in fault slip rate and precursory process along tectonic fault zones.

Concerns with main conclusions:

In my view the authors of the manuscript "Foreshock properties illuminate nucleation processes of slow and fast laboratory earthquakes" reveal some differences in AE foreshock properties from stick-slip instabilities produced in a double-direct shear apparatus depending on different normal stress levels. The main discussion and conclusion is based on repeating earthquakes which are solely determined by elevated cross-correlation coefficients, and not by location and overlapping source radii, which is in my view not sufficient in order to classify repeating earthquakes (concern 1).

Also, choosing ccc limits ≥ 0.7 for repeating earthquakes (during high normal stresses) is maybe not the best choice (concern 2) as ccc values for repeater identification in literature often exceed 0.95 (e.g., Chaves et al. (2020); Kato et al. (2016); Nadeau and Johnson (1998); Naoi et al. (2015)).

Aside of the fact, that only waveform similarity is considered as a criterion for repeating earthquakes, the authors argue that they are observing an increase in similarity after $\sim 80\%$ of the shear-stress is reached before failure for the fast SSI's, which I can to some extent support when investigating Figure 3. However, in this conclusion you are only considering AE similarities on the respective normal stress level (reference background level). According to your data, statistically, AE's exhibit higher similarities for the slow SSI's (concern 3).

Thank you for your thorough review; we greatly appreciate your comments and suggestions. We agree that waveform similarity alone is insufficient for classifying events as repeaters. In the revised text we now include additional measurements of pair-wise differential travel times, as summarized above in our reply to R1. With the additional work, we now show that by tracking differential travel times throughout the seismic cycle we can gain insights about how the events are evolving in space relative to another (Figure 4). We now show that the differential travel times evolve systematically throughout the seismic cycle and complement the waveform similarity measurements (Figure 4). We have deemphasized the idea about repeaters since our waveform similarity measurements are typically lower than 0.90. We instead focus on the relative changes in waveform similarity and differential travel times and their connection to localization (see lines 202-231). It is important to note that the absolute value of the cross-correlation will depend strongly on the time window and the frequency of the dominant signal. Our dataset is not comparable to those in natural fault systems in this regard as we are recording within frequency bands and time windows that differ by orders of magnitude from regional seismic networks.

If we refer to events that occur during the early stages of the seismic cycle (instantaneous shear stress $< 50\%$ of the peak stress) as "background" events, then it is true that the background events for small/slow slip events at 8 and 10 MPa have higher similarity compared to the large/fast events at 12 and 14 MPa. However, this is consistent with our general hypothesis that high waveform similarity is a fingerprint of fault slip. During the inter-seismic period for slow slip events, the fault never locks up and is continuously moving. In contrast, during the fast stick-

slip evens the fault locks up during the inter-seismic period and only begins to unlock and accelerate once the fault reaches ~ 80% of its peak stress. (see lines 232-240)

Significance:

The topic of understanding properties of pre-seismic fault slip certainly has very high relevance in the field of earthquake physics. However, as mentioned above, your conclusions based on repeating earthquakes lack robustness. A nice overview on repeating earthquakes was published by Uchida and Bürgmann (2019) you cite this study in your manuscript. In section 2 they nicely state the prerequisites for true repeaters. Another study you also have as a reference in your study is Gao et al. (2021). Their study “Misconception of Waveform Similarity in the Identification of Repeating Earthquakes” is focused on the problem stated above.

Because of the stated concerns, drawing main conclusions in the direction of nucleation models (L. 245-249) are made too soon in my understanding.

Thank you for your comments. See comment above about repeaters.

Suggestions:

I would suggest the authors to improve spherical coverage of AE sensors for an accurate AE location. I am not very familiar with the double-direct shear apparatus at Pennsylvania State University, but yes, I presume you have had these thoughts as well and maybe for these experiments with no confinement an improved spherical coverage is possible. Also deploying sensors which are calibrated in an absolute sense (McLaskey et al., 2015) would allow you to estimate magnitudes and corner frequencies. I am aware myself and that the authors are stating in their manuscript that event location is difficult (L. 66-70), however, to me, an accurate location is unfortunately almost a pre-requisite for any repeater study.

Once, AE sensor coverage is improved you could apply similar technics as Trugman et al. (2020), or continue with complementing your similarity study with a cluster analysis and refine arrival times per determined cluster. Use a relative location method to more accurately locate potential repeating AE's (e.g., Officer et al. (2022), Dong et al. (2019)). If absolutely calibrated AE sensors are deployed estimate source radii (e.g., Selvadurai (2019)). Unfortunately, the experiment would have to be repeated with improved AE sensor coverage and calibrated sensors.

Thank you for your comments. See comment above about repeaters and event locations. The machine, sample, and acoustic loading platens are shown in Figure 1A. Due to the geometry of the fault, our spherical coverage is limited. We have a total of 17 sensors that are located on both sides of the fault (Figure 1A), but determining absolute locations is complicated, as you note. Thus, we use pair-wise differential arrival times between event pairs which provides high resolution relative locations between events. We now show that the differential arrival times evolve systematically throughout the seismic cycle and show distinct differences for slow and fast stick-slip events. Hence, by tracking this measurement throughout the seismic cycle we can track the relative locations between event pairs.

We agree that having absolute magnitudes and corner frequencies would be beneficial, but this is beyond the scope of the current study.

Attachments:

- comments in article (Article_reviewer_comments.pdf)
- comments to figures (Reviewer_comments_on_figures.pdf)

References:

Chaves, E. J., Schwartz, S. Y., & Abercrombie, R. E. (2020). Repeating earthquakes record fault weakening and healing in areas of megathrust postseismic slip. *Science Advances*, 6(32), eaaz9317. <https://doi.org/10.1126/sciadv.aaz9317> (Science Advances)

Dong, L., Zou, W., Sun, D., Tong, X., Li, X., & Shu, W. (2019). Some developments and new insights for microseismic/acoustic emission source localization. *Shock and Vibration*, 2019.

Gao, D., Kao, H., & Wang, B. (2021). Misconception of Waveform Similarity in the Identification of Repeating Earthquakes. *Geophysical Research Letters*, e2021GL092815. (Geophysical Research Letters)

Kato, A., Fukuda, J. i., Kumazawa, T., & Nakagawa, S. (2016). Accelerated nucleation of the 2014 Iquique, Chile Mw 8.2 earthquake. *Scientific reports*, 6(1), 1-9.

McLaskey, G. C., Lockner, D. A., Kilgore, B. D., & Beeler, N. M. (2015). A robust calibration technique for acoustic emission systems based on momentum transfer from a ball drop. *Bulletin of the Seismological Society of America*, 105(1), 257-271. (Bulletin of the Seismological Society of America)

Nadeau, R. M., & Johnson, L. R. (1998). Seismological studies at Parkfield VI: Moment release rates and estimates of source parameters for small repeating earthquakes. *Bulletin of the Seismological Society of America*, 88(3), 790-814. (Bulletin of the Seismological Society of America)

Naoi, M., Nakatani, M., Igarashi, T., Otsuki, K., Yabe, Y., Kgarume, T., Murakami, O., Masakale, T., Ribeiro, L., & Ward, A. (2015). Unexpectedly frequent occurrence of very small repeating earthquakes ($-5.1 \leq MW \leq -3.6$) in a South African gold mine: implications for monitoring intraplate faults. *Journal of Geophysical Research: Solid Earth*, 120(12), 8478-8493. (Journal of Geophysical Research: Solid Earth)

Officer, T., Zhu, L., Li, Z., Yu, T., Edey, D. R., & Wang, Y. (2022). Application of the double-difference relocation method to acoustic emission events in high-pressure deformation experiments. *Physics and Chemistry of Minerals*, 49(8), 1-13.

Selvadurai, P. A. (2019). Laboratory insight into seismic estimates of energy partitioning during dynamic rupture: An observable scaling breakdown. *Journal of Geophysical Research: Solid Earth*. (Journal of Geophysical Research: Solid Earth)

Trugman, D. T., McBrearty, I. W., Bolton, D. C., Guyer, R. A., Marone, C., & Johnson, P. A. (2020). The Spatiotemporal Evolution of Granular Microslip Precursors to Laboratory

Earthquakes. Geophysical Research Letters, 47(16),
e2020GL088404. <https://doi.org/https://doi.org/10.1029/2020GL088404>

Uchida, N., & Bürgmann, R. (2019). Repeating earthquakes. Annual Review of Earth and Planetary Sciences, 47, 305-332. (Annual Review of Earth and Planetary Sciences)

L84-85. Quite a statement

Not sure what is meant here. We want simply to note that the foreshocks are driven by fault creep which is consistent with standard earthquake nucleation models/foreshock generation.

L92: Are you also having 500+ seismic cycles? I Presume its less..., please state. Also, you are investigating a single experiment here, right? State that too please.

We have over 500 seismic cycles which is equivalent to the number of lab earthquakes/stick-slip events.

L97: This information is not included in the method part.

This information is on lines 287-288.

L98: ...or maybe there are more experiments, and you just show the p5615 - unclear to me.

Data are from one experiment. This has been clarified. See line 289.

L92-103: This should go into the method section...

Done

L102: Please state to what exact location in the supporting information you are referring to?

This section has been moved to the methods section.

L108: Discrimination between slow and fast slip events at 500um/s does not correlate with Figure 1C where the discrimination seems at < 400um/s. Please clarify...

Good catch. The bifurcation is now at 500 $\mu\text{m/s}$ as stated in the text.

L109:referring once to Figure 1C here would be enough in my understanding.

Done

L114-115: ...please state where exactly in the supporting information. Could not find more information on detection.

This is a typo; the event detection information is in the methods section. See lines 320-346.

L116: What recurrence interval? Recurrence interval of similar events?

Recurrence interval of the seismic cycle. See line 100.

L119-120: Where exactly?

This is typo; details regarding waveform similarity are presented in the methods section and highlighted in Figure 2. See lines 320-346.

L124-127: Identifying repeating earthquakes solely based on waveform-similarity is usually not sufficient to distinguish between repeating and neighboring events (Gao et al., 2021) you included that article in your references too. I know that estimating accurate locations, as well as source dimensions is very challenging, especially in your case. However, the statements in your study heavily rely on true repeating earthquakes.

Good point. We have removed this statement and others that refer to events as repeaters.

L129-130: This is an important point, you compare ccc's to its background ccc on a respective normal stress level. You don't compare absolute ccc values over the entire experiment.

We removed the definition of high cross-correlation coefficients, as it was a distraction to the main points of the paper.

L140: ...for each normal stress level.

This statement has been removed.

L168-169: ...this is interpretation, should be elaborated but not in the result section.

Done.

L181-183: State references to this interpretation

Done.

L194-195: When talking about cross-correlation coefficients, you usually talk about the maximum, which means you could get ride of the "max"...

Done.

L197: Could you state what properties you are assuming here? $5500[\text{p-wave velocity, m/s}] * 2e-6[\text{differential travel times, s}] = 0.011\text{m}$

This statement has been removed. However, see lines 122-123.

L204-205: Differential travel-times are to me a side product of cross-correlations. An independent measure such as location or source size would certainly be beneficial, if not mandatory.

We now calculate pair-wise differential travel-times using arrival time data and not from cross-correlation measurements. Thus, they are not a byproduct of cross-correlation, but rather an independent and complementary measurement to waveform similarity.

L212: quartz

Not sure where the typo is here.

L216-217: Could you elaborate a bit more here? Are these fault patches which you are defining as responsible for the radiation of seismic energy now rock-rock asperities or weaknesses within the fault gouge, or maybe a mix of the

AEs could result from 2 different micro-mechanical processes. 1. Failure of asperity-asperity grain contact junctions. 2. Inter-particle slip among gouge grains (i.e., within the gouge zone). As mentioned in the text, most of the interface is covered in a thin layer of quartz powder, so it's likely that most of seismic radiation is coming from the failure inter-particle slip among gouge grains (#2). See Figure 5 and discussion section for additional details.

L220-221: You are right here, but there is throughout the stick-slip cycles at low normal stresses a higher number of more similar waveforms.

Correct. See lines 167-171.

L227: ...increase in waveform similarity compared to the background, and not overall normal stress levels.

Correct. See lines 167-171.

L228: ...laboratory seismicity in general or repeaters? May be more specific here...

We are referring to laboratory seismicity/acoustic emissions in general.

L234: ...corner frequencies of mining seismicity and micro-earthquakes are usually below ~10kHz (see e.g., Kwiatek (2011), Figure 8).

Thanks. We clarified this statement.

Kwiatek, G., Plenkers, K., Dresen, G., & Group, J. R. (2011). Source parameters of picoseismicity recorded at Mponeng deep gold mine, South Africa: Implications for scaling relations. *Bulletin of the Seismological Society of America*, 101(6), 2592-2608. (Bulletin of the Seismological Society of America)

L239-240: **...repetition and extension of the above statement,

Thanks. We removed this statement.

L257: ...until 80% of shear stress is reached

Thanks. We clarified this statement.

L264: ...75th percentile for these events is higher thorough the experiment?

The 75th percentile has been removed.

L306-307: ..why did you choose this AE sensor distribution?

At the time of the experiment our DAQ was limited to 17 sensors, and we wanted sufficient station coverage on one side of the fault so that the entire fault area was covered. Hence, we measured data from 14 stations on the left side and 3 on the right side of the DDS.

L318: How sure are you that you detected/associated real laboratory earthquakes on six stations are? What about other signals like electronic interferences, crackling of the apparatus, etc.? Location would certainly help to distinguish better!

It's unlikely that a noise spike will occur on all 17 stations during a given event. It's possible that some channels have high frequency noise that could get picked up as a false positive. However, ensuring that at least 6 stations record the same event eliminates the number of false positives. There are also several steps during the phase picking routine that help eliminate bad picks and waveforms.

L325-326: How do you deal with overlapping events?

If there are overlapping events then our detection algorithm will likely pick this up during the detection stage. These events get discarded once we extract our templates and associate the waveforms.

Figure 1:

a.)

- - Your highest normal stress level is below 14MPa, how comes and why do you state that

you have a stress level of 14MPa? This is to me not a problem per se, but you should state it accordingly.

We now clarify this in the text.

- - Stating the experiment number p5615 has no benefit for the reader, that's why I would suggest removing it.

We put experiment numbers on plots so that readers can track the origin of data and reproduce or check our results. Distributing raw data is now a common practice and using experiment numbers is an important part of reproducible science.

- The tag "Unload/reload cycles" is too prominent and distracts the reader from the important details on this figure. Removing it and stating it in the caption would be a solution.

Done

- - Time axis: Giving the time in hours referenced to the start of loading would facilitate to read this overview figure.

In all our experiments time T_0 corresponds to the start of the experiment. The onset of loading occurs when the load-point displacement is > 0 . We prefer to leave the time variable as is.

- - Drawing of the DDS:
 - Show first the 3D including the 3D coordinate axis. Then on the right show the acoustic blocks on top of each other in 2D. You could even forget about the grid, which is actually information, which is not needed, and just draw the AE's with dots on empty, colored rectangles. Give the two blocks a title, that the reader knows right away what's going on by just looking at the figure without reading the caption.
 - The fault slip sensor is not really embedded. Could you make it clearer? What do you read from that sensor? The "Load-Point Displacement"? Does the sensor need to be that prominently sketched? It draws a lot of attention.

Thanks. We prefer to leave the sketch of the DDS and acoustic blocks as is. It's a simple sketch that depicts everything the reader needs to know about the setup and location of the sensors. The fault slip sensor measure's fault slip (see lines 305-307). This is a standard sketch of the holder that mounts the slip sensor to the center block.

b.)

- - You seem to have background noise, possibly not physical, on the shear stress data, I

would consider filtering it, and estimate the stress drops from the filtered data. c.)

We filter the shear stress data prior to estimating the stress drop. However, we prefer to show the raw/unfiltered data in Figure 1.

- - Discrimination between slow and fast slip events at 500um/s stated in text does not correlate with discrimination on Figure. Discrimination seems at < 400um/s. Please clarify...

Figure 1C has been updated to account for the 500 um/s threshold.

- - Why are there five colors in the scatter plot, but only four normal stress levels stated (the yellow is missing)?

There are four colors plotting in Figure 1C each corresponding to a specific normal stress (e.g., 8,10,12,14 MPa).

d.) (I like this figure, from it, it seems that your detection/association is working nicely)

- - Maybe you could color code the waveforms, blue for right hand side AE sensors, red for left hand side AE's (you could also take out the red color to not intervene with normal stress)

We prefer to leave the figure as is. The top 3 traces coincide to the right side of the DDS and the first 14 coincide to the left side of the DDS. The reader can see this from the sketch in Fig 1A.

- - Are you band passing the waveforms? If not, please state that these are raw, unfiltered recordings.

Done

- - Shouldn't the y-axis label be "stations".

No; the y-axis is amplitude; traces are offset vertically for clarity.

Figure 2:

- - These are nice stick-slip sequences...!
- - The red color of the ccc evolution reminds of the normal stress or slip-velocity in figure

1, you could give it a turquoise color, as you color the peaks. If you like the red color, please also color the axis, as you did in Figure 1 or draw a legend. Overall, the solid line for ccc evolution is a bad choice, as the ccc is not a continuous measurement...

- - Same as in Figure 1: Reference the time to the start of loading.

Thanks. We have removed this figure in the revised ms.

Figure 3:

- State what the different CC_High are. It's always the ccc at the 75% percentile, state it directly on the plot please. Your values from Figure S2:

- 8MPa n_stress: 0.834
- 10MPa n_stress: 0.814 ○ 12MPa n_stress: 0.734 ○ ~14MPa n_stress: 0.704

This also means, that there are more events with higher ccc's for the lower normal stress levels, assuming that there is roughly an equal number of events recorded per normal stress level.

We have modified this figure and CC_High has been removed. We now use a moving median to show the relative changes in similarity throughout the seismic cycle.

Figure 4:

a.)

- - I don't really understand what you would like to say with this figure. Yes, the most cross-correlation coefficients you are looking at have small differential travel times (<2us). But you are lacking the comparison to the other group of events which exhibit smaller ccc below the 75th percentile (I presume you are looking at pairs which exhibit ccc > 75th percentile per normal stress level?).
- - Isn't this figure dominated by observations from slow SSI's? Slow SSI's have compared to fast SSI's statistically higher ccc's.

Thank you. This figure has been removed from the manuscript.

b.) and c.)

- - The same contact intensity for rock-rock asperities in figure b.) and c.) is misleading, would consider fading out the circles in b.)
- - I would reverse the colormap, as more intensity in colors usually means increasing

physical values.

Thank you for your suggestions. This figure has been modified in the revised ms.

Reviewer #3 (Remarks to the Author):

Review of manuscript NCOMMS-22-30982 submitted to Nature Communications
Foreshock properties illuminate nucleation processes of slow and fast laboratory earthquakes by David C. Bolton, Chris Marone, Demian Saffer, and Daniel T. Trugman

This is a well-presented study of stick slip tests performed on quartz gouge embedded between Westerly granite blocks in a DDS apparatus. The authors modified the normal stress to induce slow and fast slip events and monitored the AE signals during the tests. They then analyzed waveform similarity between events as function of time to failure and combined this with differential travel time analysis in an effort to estimate relative locations of AE events. These methods are well established in the analysis of field seismological data but haven't yet been applied in this way to lab tests with AE monitoring.

The authors find that waveform similarity shows significant increase towards failure for fast slip events but not for slow ones. Increase in waveform similarity is interpreted as indicating formation of localized fault patches at which slip nucleates. Since similar signatures were not observed for slow slip events, the authors posit that slow slips lack a clear nucleation stage.

Thank you for your comments and feedback.

As stated by the authors the study largely confirms the conceptual view of processes leading to stick slip failure in laboratory tests that has emerged from many previous publications which are mostly cited here. To what extent the contention is justified that a significant difference exists between processes leading to slow slip versus fast slip events is hard to tell as long as it is just based on the proxies analyzed here. Other studies suggest that event clustering is more affected by fault roughness but may occur prior to both fast and slow slips (Goebel et al., 2017, doi:10.1130/G39147.1).

Yes, we agree. It's not entirely obvious how/if our results scale up to tectonic fault zones and the implications for the slow/fast slip modes that nucleate along crustal fault zones. The revised text now includes pair-wise differential travel-time (DTT) measurements which provide information about the relative locations/distances between events. For slow stick-slip, the DTTs are roughly

constant throughout the stick-slip cycle. In contrast, the DTTs show a significant reduction prior to failure for the fastest stick-slip events at 14 MPa. The DTTs are complementary to the waveform similarity measurements; that is when waveform similarity reduces the event similarity increases, providing additional support for the coalescence of foreshocks prior to failure. Hence, DTT, waveform similarity, and temporal changes in slip rate (see supplement) all indicate that the nucleation process of slow and fast stick-slip events are distinct. See discussion for additional details.

While the results are generally in line with the current view of the nucleation process (as defined here), the study of temporal changes of AE waveform similarity is new and may well be a very valuable tool in analyzing labquake nucleation. This raises the only question or rather comment I have with this paper: The authors have an array of 17 stations at their disposal (BTW, it would be important to have a clear picture of the sensor geometry in a figure). Not knowing the set up of the sensor array I would assume this could be used to actually locate the events, which in turn would allow using hypocenter cluster analysis in space-time that could then be directly compared with the proposed waveform similarity evolution. In my view this could significantly strengthen the main results/conclusions from this work.

Thank you for comments and suggestions. The network geometry is depicted in Figure 1A. As mentioned above, we now use pair-wise differential arrival time measurements as proxies for relative locations. If two events are co-located they should produce similar move-out vectors and will have small differential arrival times. Thus, we can track relative locations between event pairs by measuring differential arrival times throughout the seismic cycle. (see lines 123-131).

In summary, this clearly is an interesting study, that deserves publishing with some minor revision. Including a comparison with evolution of AE hypocenters correlation and possibly source types would clearly strengthen the paper.

Sincerely
Georg Dresen

Thank you for your comments and feedback.

REVIEWER COMMENTS

Reviewer #1 (Remarks to the Author):

The authors made proper revisions to the paper or full reply in response to all reviewing comments. Especially for the relative spatial location of similar events that the three reviewers are concerned about, the authors added "Pair-wise differential arrival time" analysis. Under the condition that the AE event cannot be accurately located (the AE waveform were sampled at ~2Hz low), this additional work enhanced the credibility of the main conclusions of this study. I only have one concern and one suggestion:

1) The author's reply 6 (citation) is not reflected in the main text, which is related to the latest progress of using waveform cross-correlation technology to improve the detectability and location accuracy of AEs in the laboratory.

2) It is just a suggestion. The first and third rows of the upper and lower columns of Figure 5 are exactly the same. A branch diagram of 4 subplots will be simpler and clearer. It is also more consistent with the facts revealed in Figure S2-C

Reviewer #2 (Remarks to the Author):

2nd round of review of manuscript NCOMMS-22-30982A "Foreshock properties illuminate nucleation processes of slow and fast laboratory earthquakes" by David C. Bolton, Chris Marone, Demian Saffer, and Daniel T. Trugman

Dear authors, thank you for all the previous corrections! I think it's a good idea to not refer to your events as repeaters but to complement your cross-correlation-coefficient (CCC) study with pair-wise differential travel times (DTT) to circumvent an accurate absolute or relative location of AEs. You then base a lot of your discussion and conclusions on these proxies for locations. During slow stick-slip events and a continuous creeping of your lab fault, you observe high CCC's and low DTT's AE's and suggest that they are broadcasted from an ensemble of localized regions across the entire fault zone (L. 210-212). In contrast, during fast stick-slip events, the intermediate stage of a seismic cycle and a presumably locked fault, AE's are spreading out in space (L. 228 - 229) manifesting in low CCC's/ high DTT and are localized again as the fault approaches failure towards the end of the cycle (L.227).

In my understanding to make the distinction between localized and spread-out AE's you are missing the step of clustering, meaning the grouping of AE's into potential localized families (see e.g., Uchida (2019) in section "Formation of repeater sequence from pairs of repeaters"). As you do not have locations you could cluster your AE's based on your event CCC's (e.g., Shearer et al. (2003)) or/and event DTT's. (I just realized that reviewer #3 also made this suggestion in the 1st review round)

The open questions I have are the following: What do your comparable high CCC's actually mean during slow stick-slip events? Are all the AE's exhibiting high CCC's stemming from one location or is there an ensemble of locations like you describe it? It should reflect in your DTT's, right, however, statistically the two scenarios would result in the same plot as you show in Figure4AB. How many locations/groups can you identify? How do the groups of AE's exhibiting high CCC's compare to potential high CCC groups of AE's during fast stick-slip events at the initial loading stage and when approaching failure? Do all the groups of AE's disappear during the phase of 20 - 80 % of peak stress and fast stick-slip events?

Further comments:

- Abstract (L37-38): You state here that CCC's and DTT's do not evolve during the seismic cycle. I would suggest also to state that CCC's and DTT's remain at a high/low level compared to the AE's broadcasted during fast stick-slip events.

- L135: I would consider moving the explanation of DTT calculations to the method section.

- Figure 5: This figure was very hard to understand for me, could you simplify it? The message seems easy, you basically have two types of AE distributions, localized, and spread out. Confusing are these diagonal light brown shear bands. How should the reader connect them to your DDS configuration?

Minor comments:

- L122: How do you calculate fault area, under what assumptions?
- L163: "Median" waveform similarity reduces from...
- L193: Figure 4 (Figure 5) ?
- L223: On the other hand, "for fast stick-slip events" DTT ...
- L233-235: Please indicate the source of these information...
- L243: (Figure 5) \diamond to what exactly in Figure 5 are you referring to?

References

Shearer, P., Hauksson, E., Lin, G., & Kilb, D. (2003). Comprehensive waveform cross-correlation of southern California seismograms: part 2. Event locations obtained using cluster analysis. AGU Fall Meeting Abstracts,

Uchida, N. (2019). Detection of repeating earthquakes and their application in characterizing slow fault slip. *Progress in Earth and Planetary Science*, 6(1), 40. (Progress in Earth and Planetary Science)

Reviewer #3 (Remarks to the Author):

The authors have addressed my main queries and concerns and revised the manuscript accordingly. From my end the paper could no be published.

Review comments are provided verbatim below in black Times New Roman font.
Our responses are provided in green Times New Roman font.

We include a clean version of the manuscript and one showing track changes.
Line numbers quoted here are for the clean version of the manuscript.

Reviewer #1 (Remarks to the Author):

The authors made proper revisions to the paper or full reply in response to all reviewing comments. Especially for the relative spatial location of similar events that the three reviewers are concerned about, the authors added "Pair-wise differential arrival time" analysis. Under the condition that the AE event cannot be accurately located (the AE waveform were sampled at ~2Hz low), this additional work enhanced the credibility of the main conclusions of this study. I only have one concern and one suggestion:

1) The author's reply 6 (citation) is not reflected in the main text, which is related to the latest progress of using waveform cross-correlation technology to improve the detectability and location accuracy of AEs in the laboratory.

Thank you. This has been addressed in the revised text. See line 74.

2) It is just a suggestion. The first and third rows of the upper and lower columns of Figure 5 are exactly the same. A branch diagram of 4 subplots will be simpler and clearer. It is also more consistent with the facts revealed in Figure S2-C

Thank you. We have modified the figure caption to better explain our schematic. Our sketch is motivated by the data in Figures 4, S2-S4. The fast stick-slip events show three different regimes defined by low DTT/high similarity and high DTT/low similarity between 0-20, 20-80, and 80-100 % of peak stress, which we depict using three different sketches. Although the slow events only have one regime defined by low DTT and high waveform similarity, we show three different sketches for consistency with the fast stick-slip events.

Reviewer #2 (Remarks to the Author):

2nd round of review of manuscript NCOMMS-22-30982A "Foreshock properties illuminate nucleation processes of slow and fast laboratory earthquakes" by David C. Bolton, Chris Marone, Demian Saffer, and Daniel T. Trugman

Dear authors, thank you for all the previous corrections! I think it's a good idea to not refer to your events as repeaters but to complement your cross-correlation-coefficient (CCC) study with pair-wise differential travel times (DTT) to circumvent an accurate absolute or relative location of AEs. You then base a lot of your discussion and conclusions on these proxies for locations. During slow stick-slip events and a continuous creeping of your lab fault, you observe high CCC's and low DTT's AE's and suggest that they are broadcasted from an ensemble of localized regions across the entire fault zone (L. 210-212). In contrast, during fast stick-slip events, the intermediate stage of a seismic cycle and a presumably locked fault, AE's are spreading out in space (L. 228 - 229) manifesting in low CCC's/ high DTT and are localized again as the fault approaches failure towards the end of the cycle (L.227).

Thank you for your comments and feedback.

In my understanding to make the distinction between localized and spread-out AE's you are missing the step of clustering, meaning the grouping of AE's into potential localized families (see e.g., Uchida (2019) in section "Formation of repeater sequence from pairs of repeaters"). As you do not have locations you could cluster your AE's based on your event CCC's (e.g., Shearer et al. (2003)) or/and event DTT's. (I just realized that reviewer #3 also made this suggestion in the 1st review round)

This is an interesting suggestion. As described below, we have explored this line of analysis and include the results in the main text (Lines 251-279) and Figures S3-S4 in the supporting information. It is important to point out that our original waveform similarity and differential travel time measurements are computed using a moving window throughout the seismic cycle (e.g., Figure 4); therefore, the similarity measurements that we obtain are local measurements based on a subset of events and not the entire catalog. These measurements are not exactly comparable to those that are used in traditional studies to define repeaters/clusters of waveforms. Therefore, in our new analysis we performed an independent and parallel workflow that ignores the time dependence/local evolution of similarity throughout the seismic cycle and instead focuses on how/if there are distinct families of waveforms embedded with the catalog, as you suggest.

In order to define families/groups of similar AEs, we measured the pair-wise waveform similarity between all event pairs across 5 seismic cycles (Figures S3-S4). We applied a hierarchical agglomerative clustering (HAC) algorithm to cluster waveforms based upon their cross-correlation coefficients. Specifically, we clustered events using a pre-computed distance matrix of $1-C_{i,j}$, where $C_{i,j}$ is the median cross-correlation coefficient between events i and j , and applied complete linkage to merge neighboring clusters. We implemented HAC because it doesn't require the number of clusters to be determined a-priori, but instead determines the number of clusters based upon a distance threshold which is directly related to the cross-correlation coefficient. For both slow and fast stick-slip events, we used a distance threshold of 0.4 to split the dendrogram into separate clusters. The 0.4 threshold indicates that each cluster is composed of waveforms that have cross-correlation coefficients between 0.60-1. This threshold is to some extent arbitrary, and thus, the results from this analysis should be used with caution. On the other hand, because the distance matrix is calculated using waveform similarity measurements the distance threshold has a physical and meaningful definition. For example, selecting a distance threshold at 0.70 implies that all

waveforms that compose each cluster have cross-correlation coefficients 1-0.25. This threshold would permit dissimilar waveforms to exist within the same cluster, and thus, would defeat the purpose of identifying groups/clusters of similar waveforms.

We identify between 23-36 families of waveforms for slow slip cycles and between 12-15 families during fast-stick-slip cycles. However, as pointed out above, the number of clusters is strongly dependent upon the distance threshold (i.e. increasing the distance threshold above 0.4 is not ideal because clusters/families of AEs would be composed of dissimilar waveforms as noted above). Thus, the number of clusters identified at a distance threshold of 0.4 represents a lower bound on the number of potential families.

For both slow and fast stick-slip cycles, we plot the total number of families as a function of position within the seismic cycle (i.e., normalized shear stress). For slow slip events, the number of families does not evolve systematically throughout the seismic cycle. Aside from the second stick-slip cycle in Figure S4, the fast stick-slip events show a gradual increase in the number of families as co-seismic failure approaches. If one assumes that the number of families is a proxy for the number of potential locations, then the data in Figures S2-S4 indicate that during the inter-seismic period of slow slip events there are more locations across the fault plane that are failing in comparison to fast stick-slip cycles. Furthermore, the number of locations increases as failure approaches for fast-stick slip events. This is consistent with the idea that fault slip rate modulates AE activity: During slow slip the fault is continuously creeping and thus radiating events from an ensemble of locations. In contrast, during fast stick-slip the fault only begins to unlock and creep once it reaches ~ 80% of its peak strength, which could explain why the number of AE families is highest near failure.

However, we note that using the number of families as a proxy for the number of potential locations across the fault plane is only valid if the source location is the main factor that modulates waveform similarity. This neglects the possibility that two events could be co-located, have different source time functions, and thus, have waveforms that correlate poorly with one another and are a part of two different families.

In addition, we clustered waveforms using differential travel time measurements and found minor differences relative to clustering the data based on waveform similarity. The results in Figures S3 and S4 are also independent of whether all the waveforms are concatenated across all 5 seismic cycles or the waveforms from each seismic cycle are treated independently.

The open questions I have are the following: What do your comparable high CCC's actually mean during slow stick-slip events? Are all the AE's exhibiting high CCC's stemming from one location or is there an ensemble of locations like you describe it? It should reflect in your DTT's, right, however, statistically the two scenarios would result in the same plot as you show in Figure4AB. How many locations/groups can you identify? How do the groups of AE's exhibiting high CCC's compare to potential high CCC groups of AE's during fast stick-slip events at the initial loading stage and when approaching failure? Do all the groups of AE's disappear during the phase of 20 – 80 % of peak stress and fast stick-slip events?

Thank you for comments and suggestions. These are great and interesting questions. Please see comments above, Figures S3-S4, and lines 239-279.

If understood correctly, your assumption is that the number of families should be a proxy for the number of locations. However, as noted above, this is only valid if the source location is the main factor that modulates waveform similarity. This doesn't account for the possibility that two events could be co-located, have different source time functions, and thus, have distinct waveforms that correlate poorly with one another and are a part of two different families.

Further comments:

- Abstract (L37-38): You state here that CCC's and DTT's do not evolve during the seismic cycle. I would suggest also to state that CCC's and DTT's remain at a high/low level compared to the AE's broadcasted during fast stick-slip events.

Done.

- L135: I would consider moving the explanation of DTT calculations to the method section.

Done

- Figure 5: This figure was very hard to understand for me, could you simplify it? The message seems easy, you basically have two types of AE distributions, localized, and spread out. Confusing are these diagonal light brown shear bands. How should the reader connect them to your DDS configuration?

Thank you. Our sketch is motivated by the data in Figures 4, S2-S4. The fast stick-slip events show three different regimes defined by low DTT/high similarity and high DTT/low similarity between 0-20, 20-80, and 80-100 % of peak stress, which we depict using three different sketches. Although the slow events only have one regime defined by low DTT and high waveform similarity, we show three different sketches for consistency with the fast stick-slip events.

It is well known that laboratory and tectonic fault zones accommodate shear strain and deformation through shear planes (e.g., Scuderi et al., 2017). We depict these as light shades of brown that run parallel and sub-parallel to the direction shear. Because deformation is localized along shear planes it is reasonable to assume that most AEs are nucleating along these structures (See lines 190-194).

Minor comments:

- L122: How do you calculate fault area, under what assumptions?

The fault area is determined by the dimensions of the side blocks that make up the double-direct shear configuration (see Figure 1 and methods section). The blocks used in this experiment have a nominal contact area of 50 x 50 mm². The only assumption is that the fault zone retains a finite thickness –and does not break into segments separated by nothing—and this is verified by post-

run observations that are done in every experiment. The fault zones thin with shear due to compaction and geometric thinning but this is well understood (e.g., Scott et al., 1994; Kaproth and Marone, 2014) and unrelated to area.

- L163: “Median” waveform similarity reduces from...

The data in Figure 4 represent the maximum waveform similarity. This is described in lines 98-106.

- L193: Figure 4 (Figure 5) ?

Fixed.

- L223: On the other hand, “for fast stick-slip events” DTT ...

Fixed.

- L233-235: Please indicate the source of these information...

This statement is based on our data and observations. For example, during the inter-seismic period for slow slip events the fault is continuously creeping and AEs that are broadcasted have low DTT and high similarity. During fast events the fault starts to creep once it surpasses ~ 80% of its peak strength, which coincides where DTT begins to reduce and waveform similarity increases. Hence, when the fault is creeping the differential travel-times are low and waveform similarity is high.

- L243: (Figure 5) \diamond to what exactly in Figure 5 are you referring to?

This is a typo. We are referring to Figure 4 in this statement.

References

Kaproth, B. M., and C. Marone (2014), Evolution of elastic wave speed during shear-induced damage and healing within laboratory fault zones, *J. Geophys. Res. Solid Earth*, 119, 4821–4840, doi:10.1002/2014JB011051.

Scott, D. R., C. J. Marone, and G. C. Sammis (1994), The apparent friction of granular fault gauge in sheared layers, *J. Geophys. Res.*, 99(B4), 7231–7231, doi:10.1029/93JB03361.

Shearer, P., Hauksson, E., Lin, G., & Kilb, D. (2003). Comprehensive waveform cross-correlation of southern California seismograms: part 2. Event locations obtained using cluster analysis. AGU Fall Meeting Abstracts,

Uchida, N. (2019). Detection of repeating earthquakes and their application in characterizing slow fault slip. *Progress in Earth and Planetary Science*, 6(1), 40. (Progress in Earth and Planetary Science)

Reviewer #3 (Remarks to the Author):

The authors have addressed my main queries and concerns and revised the manuscript accordingly. From my end the paper could now be published.

Thank you for your feedback and comments.

REVIEWERS' COMMENTS

Reviewer #1 (Remarks to the Author):

This paper presents a research study that utilized a dataset of low-sampling-frequency AE waveform data monitored at 17 positions, as well as direct measurement data such as slip velocity. The study analyzed the evolutionary characteristics of AE events during slow and fast slip events controlled by normal stress. Under these AE observation conditions, based on my experience, high cross-correlation coefficients (CCC, greater than 0.85) and low differential travel time (DTT, less than 1 microsecond) are considered good indicators of mechanistically similar and spatially close AE events. Indeed, it cannot be confirmed that all of these events are repeated events in a strict sense. In seismology, repeated events are primarily used to assess the slip velocity of faults during slow slip events. Since fault slip velocity is an observed data in this study, it is not necessary to determine strictly repeated events. I think that the conclusions and inferences of this study, based on CCC and DTT data, are well-founded. After performing cluster analysis, the concerns raised by the second reviewer have also been satisfactorily addressed. I thus recommend publishing the study.

Review comments are provided verbatim below in black Times New Roman font.
Our responses are provided in green Times New Roman font.

**We include a clean version of the manuscript and one showing track changes.
Line numbers quoted here are for the clean version of the manuscript.**

Reviewer #1 (Remarks to the Author):

This paper presents a research study that utilized a dataset of low-sampling-frequency AE waveform data monitored at 17 positions, as well as direct measurement data such as slip velocity. The study analyzed the evolutionary characteristics of AE events during slow and fast slip events controlled by normal stress. Under these AE observation conditions, based on my experience, high cross-correlation coefficients (CCC, greater than 0.85) and low differential travel time (DTT, less than 1 microsecond) are considered good indicators of mechanistically similar and spatially close AE events. Indeed, it cannot be confirmed that all of these events are repeated events in a strict sense. In seismology, repeated events are primarily used to assess the slip velocity of faults during slow slip events. Since fault slip velocity is an observed data in this study, it is not necessary to determine strictly repeated events. I think that the conclusions and inferences of this study, based on CCC and DTT data, are well-founded. After performing cluster analysis, the concerns raised by the second reviewer have also been satisfactorily addressed. I thus recommend publishing the study.

Thank you for your feedback and comments. Yes, we agree with your interpretation about repeaters.